

# Characteristics of lower stratospheric transport as inferred from the age of air spectrum

F. Ploeger[1] and T. Birner[2]

[1]Institute for Energy and Climate Research: Stratosphere (IEK–7), Forschungszentrum Jülich, Jülich, Germany.
[2]Department of Atmospheric Science, Colorado State University, Fort Collins, CO, USA.

*Correspondence to:* XXX

**Abstract.** Trace gas transport in the lower stratosphere is investigated by analyzing seasonal and inter-annual variations of the age of air spectrum – the probability distribution of stratospheric transit times. Age spectra are obtained using the Lagrangian transport model CLaMS driven by ERA-Interim winds, and using a boundary-impulse-response method based on multiple tracer pulses. Seasonal age spectra show large deviations from an idealized stationary unimodal shape. Multiple modes emerge in the spectrum throughout the stratosphere, strongest at high latitudes, caused by the interplay of seasonally varying tropical upward mass flux, stratospheric transport barriers and recirculation. The annual mean spectrum, on the other hand, is found to be well described by an idealized stationary spectrum. Inter-annual variations in transport (e.g., QBO, ENSO) cause significant modulations of the age spectrum shape. In fact, one particular QBO phase may determine the spectrum's mode during the following 2-3 years. Interpretation of the age spectrum in terms of transport contributions due to the residual circulation and mixing is generally not straightforward. Advection by the residual circulation turns out to represent the dominant pathway in the deep tropics and in the winter hemisphere extratropics above 500K, controlling the modal age in these regions. In contrast, in the summer hemisphere, particularly in the lowermost stratosphere, mixing represents the most probable pathway controlling the modal age. Analysis of the full age spectrum compared to mean age is highly beneficial for separating the effects of different transport processes, and is strongly recommended as a diagnostic for model inter-comparisons.

## 1 Introduction

The composition of the lower stratosphere includes radiatively active trace gases such as water vapor and ozone, and these strongly affect Earth's radiation budget and surface temperatures (e.g., Riese et al., 2012; Solomon et al., 2010). The trace gas distribution in this region is strongly shaped by the global-scale Brewer-Dobson circulation, which may be separated into a residual mean meridional mass circulation and additional eddy mixing (e.g., Holton et al., 1995; Butchart, 2014). Both the residual circulation and mixing are largely driven by breaking Rossby waves and, to some extent, gravity waves (e.g., Haynes et al., 1991; Haynes and Shuckburgh, 2000).

A commonly used diagnostic to study transport in the lower stratosphere is the mean age of air, the average transit time of a stratospheric air parcel since entering the stratosphere (e.g., Hall and Plumb, 1994; Waugh and Hall, 2002). However, stratospheric air parcels are affected by various mixing processes along their pathways. Hence, an air parcel consists of a





mixture of air with different transit times and is more fully characterized by a transit time distribution, commonly referred to as the *age spectrum*. Mean age of air (the first moment of the age spectrum) only provides a succinct description of stratospheric transport for narrow and nearly symmetric age spectra. Generally, the effects of different transport processes (e.g., residual mean mass transport and mixing) cannot be distinguished with a single measure such as mean age and may lead to puzzling

results.

One such puzzling result concerns potential changes in stratospheric transport in a changing climate. Climate models indicate a strengthening residual circulation during recent decades and into the future, whereas observations of mean age show no decrease of the mean transit time (e.g., Engel et al., 2009; Stiller et al., 2012; Butchart et al., 2010). Recently, progress has been made to reconcile these apparently contradictory results, indicating a long-term strengthening residual circulation

(Abalos et al., 2015) causing decreasing mean age, superposed by strong decadal variations and a significant effect of eddy mixing (Ploeger et al., 2015a). When dividing the Brewer-Dobson circulation into a shallow and a deep branch, evidence for a strengthening circulation is found for the shallow branch from both models (e.g., Garny et al., 2011) and in-situ observations (Bönisch et al., 2011; *Ray et al.*, 2010; Ray et al., 2014).

Consideration of the full age spectrum compared to just the mean age comes at the benefit of allowing to separate the

effects of different transport processes and may ultimately lead to an improved understanding of stratospheric transport and its long-term changes. Many studies on stratospheric age spectra are based on the assumption of stationary atmospheric flow and approximate the age spectrum by the Green's function for the diffusion process (an inverse Gaussian distribution), motivated by the pioneering work of Hall and Plumb (1994). However, this is a strong simplification as stratospheric transport shows variations on multiple time scales (e.g., seasonal, inter-annual) and is clearly non-stationary (e.g., Haine et al., 2008).

Some recent studies considered non-stationary age spectra within models and presented estimates of seasonal age variations (Reithmeier et al., 2007; Li et al., 2012a; Diallo et al., 2012) and long-term trends (Li et al., 2012b). These studies found significant differences of seasonal age spectra from the idealized stationary shape, with the age spectrum within particular regions even showing multiple peaks (modes). However, their explanations for these seasonal variations and the multiple peaks are different, involving seasonality in tropical upward mass flux, the strength of the polar vortex transport barrier, or the existence

of different circulation branches. A common understanding of the intricate age spectrum characteristics appears to be lacking.

Regarding inter-annual age spectrum variations, to our knowledge the only published results (Li et al., 2012a) are based on a model simulation without a quasi-biennial oscillation (QBO). Hence, age spectrum modulations by the QBO have so far not been studied. Moreover, the common view that the age spectrum peak (*modal age*), which by definition corresponds to the most probable transit time, can be related to the residual mean mass circulation appears questionable. In several regions of the

stratosphere, mixing processes have a predominant effect on transport (e.g., in the NH lower stratosphere during summer, see Konopka et al., 2015) and likely represent the most probable transport pathway.

In this paper, we present a method to calculate stratospheric age spectra in the Chemical Lagrangian Model of the Stratosphere (CLaMS), a state-of-the-art reanalysis-driven chemistry transport model. This method is a further development of the approach by Li et al. (2012a), based on boundary impulse responses using multiple pulses launched at the Earth's surface in

the deep tropics, and allows calculating the age spectrum in a transient simulation without additional assumptions such as the





smallness of inter-annual variability. The analyses focus on the following questions: (i) How large is the variability of lower stratospheric age spectra from seasonal to inter-annual time scales and how do multiple spectral peaks (modes) develop? (ii) How do residual circulation and mixing affect the age spectrum? (iii) How accurate are commonly used approximations of the age spectrum shape?

We describe our methodology in Sect. 2 and consider seasonal age spectrum variations in Sect. 3. The effects of residual circulation transport and mixing are investigated in Sect. 4. Inter-annual variability is discussed in Sect. 5. A discussion of the development of multiple modes in the spectrum, the benefit of consideration of the full spectrum compared to mean age alone, and the goodness of commonly used age spectrum fits is presented in Sect. 6. The final section concludes the paper.

## 2 Methodology

### 2.1 The CLaMS model simulation

The model used in this study is the Chemical Lagrangian model of the Stratosphere (CLaMS). CLaMS is a Lagrangian transport model with trace gas transport based on the motion of three-dimensional forward trajectories and an additional parameterization of small-scale atmospheric mixing (e.g., McKenna et al., 2002; Konopka et al., 2004). This mixing parameterization induces strong mixing in regions of large flow deformations. Vertical transport in CLaMS is based on a hybrid $\sigma-\theta$ coordinate, which

transforms smoothly from an orography-following coordinate ($\sigma = p/p_s$, with $p$ pressure and $p_s$ surface pressure) near the surface into potential temperature $\theta$ (see also Mahowald et al., 2002). Above $\sigma = 0.3$, hence throughout the stratosphere and the TTL, the hybrid vertical coordinate exactly equals potential temperature and the vertical velocity is determined by the total diabatic heating rate. Further details about this particular model set-up are described in Pommrich et al. (2014).

    For this study, we carried out a simulation for the 1979–2013 period, with model transport driven by European Centre for

Medium-Range Weather Forecasts (ECMWF) interim Reanalysis (ERA-Interim) winds (e.g., Dee et al., 2011). Furthermore, we implemented a method to calculate the age of air spectrum within the model, which will be described in detail next.

### 2.2 Age spectrum calculation

The solution to the continuity equation for a conserved and passive tracer with mixing ratio $\chi$ at location $r$ and time $t$ may be expressed as (e.g., Waugh and Hall, 2002):

$$\chi(r,t) = \int_0^\infty d\tau \, \chi(\Omega, t-\tau) G(r,t \,|\, \Omega, t-\tau) \,. \tag{1}$$

The kernel $G(r,t \,|\, \Omega, t-\tau)$ is called the *boundary propagator*. This propagator is related to the transport operator's Green's function (see Holzer and Hall, 2000), and relates the tracer mixing ratio at $(r,t)$ to the mixing ratio in the boundary source region $\Omega$ (chosen to be the tropical boundary layer, in the following) a time $\tau = t - t'$ ago. Here, $t$ is called the *field time* when the tracer mixing ratio is sampled, whereas $t'$ is called the *source time* when the tracer had last contact with $\Omega$. Interpreting

Eq. (1), $G(r,t \,|\, \Omega, t-\tau)d\tau$ is the mass fraction of air at $(r,t)$ that had last been in contact with $\Omega$ between $\tau$ and $\tau + d\tau$ ago.





For this reason, $G(r, t \,|\, \Omega, t - \tau)$ is a transit time distribution and has been termed the *age spectrum* by Hall and Plumb (1994). Note that the propagator $G$ can be interpreted as a joint air mass origin and transit time distribution, such that integration over $\tau$ yields the mass fraction of air originating from $\Omega$ (e.g., Orbe et al., 2013, 2015).

For an inert tracer with a pulse in $\Omega$ at source time $t' = t'_0$ the source time history is given by $\chi(\Omega, t') = \delta(t' - t'_0)$, and Eq.
(1) reduces to (recalling $t' = t - \tau$)

$$\chi(r, \tau + t'_0) = G(r, \tau + t'_0 \,|\, \Omega, t'_0). \tag{2}$$

Here, $G(r, \tau + t'_0 \,|\, \Omega, t'_0)$ is the *boundary impulse response* (BIR), the time-evolving response at location $r$ to a $\delta$-distribution boundary condition in $\Omega$ at time $t'_0$. In general, as a function of transit time $\tau$ the BIR is not equal to the age spectrum, because the $\tau$-dependency occurs in different arguments of $G$. Only for stationary flow the boundary propagator is time translation invariant and depends only on the transit time $\tau$, such that the age spectrum and BIR are equal (e.g., Haine et al., 2008).

Equation (2) provides a method to calculate the boundary propagator $G$ by using $N$ different pulse tracers $\chi_i$ ($i = 1, ..., N$), with pulses in $\Omega$ at times $t'_i$ (e.g., Haine et al., 2008). The age spectrum, as a function of $\tau$, may be constructed from these $N$ pulse tracers at each field time $t$ and location $r$ as

$$G(r, t \,|\, \Omega, t - \tau_i) = \chi_i(r, t). \tag{3}$$

Hence, from $N$ pulse tracers $N$ pieces of information of the age spectrum at the discrete transit times $\tau_i$ may be deduced. Recently, this BIR-method was used to investigate the seasonality (Li et al., 2012a) and long-term behaviour (Li et al., 2012b) of stratospheric age spectra in the Goddard Earth Observing System Chemistry-Climate Model (GEOSCCM).

Here, we use a total of $N = 60$ different boundary pulse tracers, with pulses released in the lowest model layer (orography following) in the tropics between 15°S–15°N, constituting the source region $\Omega$. This source region has been chosen similar to that of Li et al. (2012a) for ease of comparison. Variations of the source region (e.g., 10°S–10°N, entire global surface layer) lead to qualitatively the same results. Note that another common choice, particularly for observationally based age spectrum estimates, is to define $\Omega$ as the tropical tropopause. Related differences for transit times are expected to be a few weeks to months, which is the time scale for transport from the surface to the tropopause. For each pulse, the particular tracer mixing ratio is set to unity in $\Omega$ for 30 days, and is set to zero in $\Omega$ otherwise. Pulses are launched every other month. Consequently, for the considered period 1979–2013 the first tracer pulse has source times in January 1979, the second tracer pulse in March 1979, and so on.

The implementation of the method in the transport model CLaMS is illustrated in Fig. 1, which shows the BIR map at 400 K and 60°N, i.e. the propagator $G$ versus field time and source time. Following the initial pulse at source time $t'$, each pulse tracer mixing ratio evolves with field time $t$ producing a vertical section in the BIR map. The set of all tracers comprises the BIR map. Therefore, a vertical cut through the BIR map provides the boundary impulse response at a given source time $t'$ as a function of field time $t$ (e.g., the vertical dashed arrow shows the BIR for January 1995). A horizontal cut through the BIR map provides the age spectrum at a given field time as a function of source time, or equivalently of transit time by noting that $\tau = t - t'$ (e.g., the horizontal arrow shows the age spectrum for June 2002).





Due to the finite number of tracers used here ($N = 60$) initialized every two months in source time (i.e. 6 pulses per year for 10 years), the calculated age spectrum is truncated to a maximum transit time of ten years. We used results of a ten year long perpetual simulation (repeating 1979 winds) as initialization for the transient simulation starting at 1 January 1979. Therefore, values in the BIR map at source times before 1 January 1979 (and field times before 1989) will be influenced by this

initialization procedure. For this reason, we do not consider age spectra before 1989 in this paper, and calculate climatologies for the 1989–2013 period. Once all sixty tracers have been used (after the first ten years) the first tracer is reset to zero and then used for the pulse at source time January 1989, and so on for consecutive pulses. Hence, we obtain time-dependent age spectra at every day of the transient 1979–2013 simulation, with a resolution along the transit time axis of 2 months. Li et al. (2012a) showed that a 2-month resolution of the pulses is suitable for resolving the seasonal age spectrum variations, and it is therefore

sufficient for the results presented in this paper.

Our implementation of the age spectrum method into CLaMS differs from the BIR calculation by Li et al. (2012a) and Li et al. (2012b), who considered time slice simulations and used each tracer only once. To emphasize the fact that the modified BIR tracer setting in CLaMS evolves with time in a transient simulation, we denote the age spectrum calculation as implemented here the *Boundary Impulse Evolving Response* (BIER) approach.

Figure 1b shows the annual mean age spectrum at 400 K in NH mid-latitudes at 60°N, the horizontal cut through the BIR map in Fig. 1a. In general, stratospheric age spectra are characterized by skewed distributions with a long tail at old ages. The maximum of the spectrum (*modal age*, here highlighted as vertical black dashed line) corresponds to the most probable transit time and likely to the most probable pathway (to be discussed further below). The *mean age* (vertical black solid line) is defined as the first moment of the age spectrum

$$\Gamma(r, t) = \int_0^\infty d\tau\, \tau\, G(r, t \,|\, \Omega, t - \tau) \tag{4}$$

and strongly depends on the tail of the distribution. Another quantity characterizing the distribution is the age spectrum width

$$\Delta(r, t) = \sqrt{\frac{1}{2} \int_0^\infty d\tau\, [\tau - \Gamma(r, t)]^2\, G(r, t \,|\, \Omega, t - \tau)}. \tag{5}$$

In addition, the red dashed line shows the residual circulation transit time (RCTT), the hypothetical transit time of an air parcel if it was advected by the residual circulation only (e.g., Rosenlof, 1995; Birner and Bönisch, 2011). RCTT's and their relation

to mean age of air have recently been discussed by Garny et al. (2014). Here, RCTT's are calculated from 2D CLaMS backward trajectories in the latitude-potential temperature plane, driven by the ERA-Interim residual circulation in isentropic coordinates $(\overline{v}^*, \overline{Q}^*)$, as described by Ploeger et al. (2015b). In this isentropic zonal mean formulation $\overline{v}^* = (\overline{\sigma v})/\overline{\sigma}$ and $\overline{Q}^* = (\overline{\sigma Q})/\overline{\sigma}$ are the mass-weighted zonal mean meridional and vertical velocities, where $\dot{\theta} = Q$ is the cross-isentropic vertical velocity and $\sigma = -g^{-1} \partial_\theta p$ is the density in isentropic coordinates, with $p$ pressure and $g$ the acceleration due to gravity (e.g., Andrews et al.,

1987, Chapter 9). The RCTTs are calculated with respect to the 340 K surface. This causes a weak young bias of the RCTTs compared to age of air, corresponding to the transit time from the Earth's surface to 340 K. The relation between the age spectrum and the RCTT will be studied in Sect. 4.



A limitation of the described approach to calculate the age spectrum originates from the limited number of pulse tracers in the model, such that only the first ten years of the spectrum can be calculated explicitly. Because the mean of the distribution strongly depends on the spectrum's tail this leads to an underestimation of mean age. However, for transit times above about 4-5 years age spectra are generally found to decay roughly exponentially (cf. Fig. 1c). The corresponding decay rate is related to the exponential decay of the mixing ratio of a conserved tracer in the stratosphere (e.g., Ehhalt et al., 2004). To estimate the age spectrum's tail for transit times larger than ten years, the spectrum may therefore be extrapolated using an exponential fit based on the values at transit times between 5-10 years (red dashed line in Fig. 1c), as explained in detail in Appendix A. A similar tail correction procedure was employed by Diallo et al. (2012). Throughout this paper we present the tail-corrected mean age.

Global mean age distributions for NH winter (DJF) and summer (JJA), as obtained from age spectra, are shown in Fig. 2. A stronger Brewer-Dobson circulation during NH winter causes younger ages in the tropics, compared to the boreal summer season. Mean age is older in the winter than in the summer extratropics, due to stronger Brewer-Dobson circulation downwelling in the winter hemisphere and weaker transport barriers in the summer hemisphere. Particularly young mean age is found in the NH lower stratosphere during summer as a result of strong quasi-horizontal mixing during this season (Konopka et al., 2015). The effect of the correction for the finite spectrum tail is an increase of mean age of about half a year compared to the uncorrected ages, mainly at high altitudes and latitudes (see Appendix A for details). Comparison to mean age calculated from a model 'clock-tracer', an inert tracer with a linear increase in the boundary layer (see also Hall and Plumb, 1994), shows good agreement with the spectrum-based mean age and therefore affirms the internal consistency of the age spectrum calculation in the model (see Fig. 2 c/d).

## 3   Seasonality of age spectra

The climatological annual mean age spectrum in the NH lower stratosphere (here 400 K, 60°N) from the CLaMS simulation (black line in Fig. 3) is characterized by a skewed distribution with a maximum around half a year. This annual mean distribution is well approximated by the idealized age spectrum for stationary flow (the Green's function for diffusion, grey shading), as suggested by Hall and Plumb (1994). This approximation will be further discussed in Sect. 6.3. The seasonal age spectra, however, show a more complicated structure with multiple peaks along the transit time axis (coloured lines in Fig. 3). The location of the global maximum in the seasonal age distribution (the modal age) depends on the season, coinciding with the youngest peak from spring to fall and with the second youngest peak in winter. Furthermore, the location of the mode strongly depends on the region, as will be shown in later sections.

The occurrence of multiple peaks in the seasonal age spectrum (Fig. 3) reflects the seasonality of transport. The highest fraction of young air in the NH lower stratosphere at 400 K is found in NH summer (red line in Fig. 3). The BIR map in Fig. 1a shows that these air masses had left the boundary layer about half a year earlier, in the previous winter (source time of the peak corresponds to NH winter). Hence, the highest fraction of young air in the NH lower stratosphere during summer is caused by particularly efficient and fast transport from the boundary layer into the stratosphere during NH winter and spring, consistent




with the results of Reithmeier et al. (2007). This efficient transport is related to strongest tropical upward mass flux in boreal winter and weakening subtropical and polar transport barriers in spring and summer.

Figure 3 further shows how the summertime peak of young air (transit times of about half a year) ages throughout the course of the year. The coloured crosses in the figure illustrate the position of the peak during the following seasons: at about 0.75 years during the following fall, $\sim 1$ year during the following winter, $\sim 1.25$ years during the following spring, with a $\sim$ 1-year offset from the original summer peak during the following year's summer, and so on. The evolution of the age spectrum peaks becomes even clearer in a presentation versus season and transit time in Fig. 4. An exceptionally high fraction of young air arrives in the extratropical NH lower stratosphere during spring and summer, causing the strong summertime peak at young transit times. This peak propagates to older transit times during the following months, and occurs around transit times of one year constituting the second peak of the spectrum during next winter. The occurrence of multiple peaks in the age spectrum is therefore directly linked to the seasonality of transport, with most efficient transport from the boundary layer into the stratosphere during NH winter. Further discussion of the spectrum peaks will be presented in Sect. 6.1.

Consideration of the full age spectrum allows quantifying air mass fractions corresponding to certain transit times. Of particular importance for short-lived chemical species in the lower stratosphere is the fraction of very young air with transit times below some threshold $\tau^*$ (denoted *fresh-air-fraction* $F_{\tau^*}$ in the following), which can be deduced from the age spectrum by integration over transit time

$$F_{\tau^*}(r,t) = \int_0^{\tau^*} d\tau\, G(r,t\,|\,\Omega, t-\tau). \tag{6}$$

Figure 5 shows the seasonality of the fraction of fresh air with transit times younger than 6 months ($F_6$) in the lower stratosphere. Clearly, the seasonality in $F_6$ is very different in the two hemispheres and in the tropics. In the tropical lower stratosphere, the highest fraction of young air is found in NH winter and spring whereas the lowest fraction is found during summer and fall, consistent with the annual cycle in tropical upwelling which maximizes in NH winter. A remarkable fraction of tropical air above about 380 K shows transit times larger than 6 months (typically more than 50%). This likely points at a significant impact of in-mixing of old air masses from the extratropics on the tropical composition. This fraction of old air in the tropics maximizes in NH summer (e.g., about 60% at 400 K), coinciding with strongest eddy mixing into the tropics during summer (e.g., Konopka et al., 2015).

The two hemispheres show opposite annual cycles in $F_6$, with maximum fractions of young air in the NH during June–October (depending on altitude) and in the SH during February–April. Note that at lower levels the $F_6$ in the NH extratropics peaks later in the year (e.g., during fall on the 340 K isentrope). The timing of the youngest air in the extratropics is consistent with strongest horizontal transport from the tropics during each hemisphere's summer. Furthermore, the approximate annual cycle in the tropical $F_6$ with minimum during NH summer is consistent with stronger horizontal eddy mixing in the NH than in the SH (e.g., Konopka et al., 2015).





## 4   Residual circulation and mixing effects on age spectra

From a conceptual point of view, zonal mean stratospheric transport may be separated into net mass transport (given by the residual mean meridional circulation) and additional two-way mixing due to eddies (e.g., Andrews et al., 1987, chapter 9). The transit time corresponding to the residual circulation, the RCTT (see Sect. 2), describes the pure effect of residual circulation transport (e.g., Birner and Bönisch, 2011). Therefore, the difference between the RCTT and the "real" atmospheric transport time scale (the age of air) provides a measure of the effect of eddy mixing (Garny et al., 2014). The peak of the age spectrum (modal age) is related to the most probable transport time scale, likely corresponding to the most probable transport pathway. Comparison between the modal age and the RCTT therefore allows an analysis of the regions and seasons where either the residual circulation or eddy mixing dominates stratospheric transport.

Figure 6 presents the age spectra at 400 K and 600 K for all latitudes during winter and summer, together with the corresponding modal ages, mean ages, and RCTTs. In the tropics, the age spectrum generally shows one distinct peak, hence a well-defined modal age, at time scales close to the RCTT. From the subtropics to high latitudes, however, the spectrum shape is characterized by multiple peaks of similar strength, and therefore the modal age is ill-defined. The annual cycle in tropical upwelling, with faster upwelling in NH winter compared to summer, is reflected in a younger tropical spectrum peak during winter. The small difference between the modal age and the RCTT in the lower tropical stratosphere at 400 K, with the modal age slightly older, shows an effect of in-mixing of old extratropical air into the tropics just above the tropopause.

At 600 K there is a clear separation of youngest air (around 1 year) in the tropics during both seasons, consistent with a tropical pipe model of stratospheric transport (Plumb, 1996). Interestingly, the modal age during NH summer appears at somewhat older transit times, suggesting that recirculation of older air from the extratropics plays a dominant role during this season.

The isolation of tropical air through subtropical transport barriers does not extend down to 400 K (e.g., Volk et al., 1996). However, a steep latitudinal gradient in the fraction of youngest air (less than 6 months) still exists in mid-latitudes at this lower level, particularly during winter in both hemispheres. This latitudinal gradient is significantly weaker during summer, in particular in the NH lower stratosphere where the high fraction of young air at high latitudes indicates strong horizontal transport from the tropics. The large difference between modal age and RCTT in this region (poleward of about 50°N during NH summer) further indicates that transport is predominantly due to mixing. On the other hand, equatorward of about 50°N modal age and RCTT agree well at both levels and seasons, which points to a more dominant role of residual circulation transport. Therefore, the age spectrum analysis is consistent with recently published results relating the summertime "flushing" of the NH extratropics with young air to the combined effect of residual circulation transport equatorward of about 50°N, and quasi-horizontal mixing poleward of about 50°N (Bönisch et al., 2009; Konopka et al., 2015). In general, the existence of multiple spectral peaks in the extratropics questions the value of modal age as a descriptor for age of air.

Profiles of modal age and RCTT at different latitude bands for winter and summer are compared in Fig. 7. Throughout the tropics, the modal age agrees well with the RCTT and consequently transport is largely related to the residual circulation. The figure also includes a simple proxy for the time scale of tropical upward transport (only panels a and d) based on the definition



of cross-isentropic vertical velocity $\dot{\theta} = \Delta\theta/\Delta t$. Note that this simple proxy significantly differs from both the modal age and the RCTT during NH summer above about 650 K. This is related to seasonal changes in the structure of tropical upwelling and apparently emerges at transit times above about 1.5 years. The wintertime high-latitude stratosphere above about 500 K (55–85°N during DJF, and 85–55°S during JJA) represents another case where modal age and RCTT closely match and transport appears to be well described by the residual circulation. For all other regions and seasons modal age and RCTT differ, at least partly, and eddy mixing has a significant effect on stratospheric transport. The clearest difference between modal age and RCTT and therefore the strongest mixing effect emerges in the high-latitude (55–85°N/S) lower stratosphere below about 500 K, in particular during summertime. However, apart from this exception the RCTT represents a surprisingly good approximation to modal age, but not mean age.

The comparison between modal age and RCTT is summarized in Fig. 8, which shows the percentage difference between RCTT and modal age (RCTT − mode)/RCTT for winter and summer. Small differences indicate a strong effect of the residual circulation on transport, as found throughout the deep tropics and in the respective winter hemisphere. Furthermore, the dipole pattern in each hemisphere below about 500 K, with RCTT smaller than the modal age in the tropics and larger than the modal age poleward of about 50°, is related to the strong effect of eddy mixing at these levels. As eddies mix air quasi-horizontally between the tropics and the extratropics relatively old air is transported into the tropics, while young air is transported into the extratropics. Therefore, in the tropics mixing causes an older modal age as compared to the pure residual circulation effect, reflected in the negative values in that region (Fig. 8). In the extratropics, mixing causes the modal age to be younger than resulting from pure residual circulation advection, and hence positive differences.

## 5 Inter-annual variability of age spectrum

Figure 9 (left panels) shows the full time series of simulated age spectra in the lower stratosphere at 600 K for the tropics and extratropics of both hemispheres. The right panels show the corresponding deseasonalized anomalies, constructed by subtracting the mean annual cycle at each transit time. Mean age, modal age, and RCTT (and their respective deseasonalized anomalies) are indicated.

The tropical spectra (Fig. 9a–b) at 600 K show strong inter-annual variability. Comparison to the tropical zonal mean zonal wind (red bars highlight easterly wind periods) implies the QBO as the main driver of these variations. During QBO easterly phases the fraction of young air increases and the modal age shifts to younger transit times consistent with enhanced tropical upwelling as expected from the QBO-induced secondary meridional circulation (e.g., Baldwin et al., 2001). Opposite behaviour is observed during QBO westerly phases. Furthermore, the modal age varies between one and two years, with inter-annual variations closely matching the variability in RCTT. This highlights the dominant influence of the residual circulation on transport variations in the tropical lower stratosphere on inter-annual time scales. Even the inter-annual variations of mean age closely follow those of the RCTT (Fig. 9b). The discontinuity in the tropical age spectrum time series at the end of 1993 (most strongly evident in panel b) could hint at problems with the reanalysis data, but we were not able to relate it to specific changes in the assimilation system.





The extratropical age spectra also show strong QBO-related inter-annual variations in both hemispheres (Fig. 9c–f). The fraction of young air increases at the end of QBO easterly phases, lagging the tropical signal by a few months. Remarkably, these young air peaks originating in QBO easterly phases propagate to older transit times and determine the modal age during the following 2–3 years. For example, the modal age during 1996–1999 is set in the easterly QBO phase in 1996 (see Fig. 9c). Analogous behaviour results from QBO westerly phases, with anomalously old air originating in the tropics and being transported poleward. The propagation of the modal age along the transit time axis clearly shows how the QBO affects the composition of the lower stratosphere on a global scale.

In the lower stratosphere (at 400 K, Fig. 10) time series of age spectra are dominated by the annual cycle of transport in this region (see Sect. 3). Modal age and RCTT show a general offset of about 1–2 months, which points to a significant effect of mixing in this region (see also Sect. 4). However, the deseasonalized time series (Fig. 10, right column), show remaining variability, which particularly in the tropics appears to co-vary partly with the multivariate El Niño–Southern Oscillation (ENSO) index MEI (Wolter and Timlin, 1993, 1998). El Niño phases are indicated in Fig. 10 by highlighting periods with a sufficiently positive MEI (here $> 0.5$). The fraction of young air masses increases during El Niño phases (positive multivariate ENSO index), consistent with enhanced tropical upwelling during these periods (e.g., Randel et al., 2009; Calvo et al., 2010). Analogous behaviour with oppositely signed anomalies is found for La Niña phases. The age spectrum therefore shifts to younger transit times during El Niño and to older transit times during La Niña. This is especially evident during strong events, such as the 1997/1998 El Niño followed by two strong La Niña years. These co-variations between the age spectrum time series and ENSO at 400 K are however generally not as succinct as the co-variations with the QBO at 600 K (see Fig. 9). Nevertheless, some of the tropical variability at 400 K emerges at middle and high latitudes (e.g., after 1998 in Fig. 10d/f), indicating a global impact of ENSO on the inter-annual variability of lower stratospheric transport.

## 6 Discussion

In this section, we first discuss the generation of multiple peaks in lower stratospheric age spectra, followed by a discussion of the additional information contained in the full age spectrum (e.g., fractions of young air) as compared to the commonly considered mean age. Furthermore, we discuss the accuracy of simple Green's function approximations of stratospheric age spectra.

### 6.1 Generation of multiple peaks

As mentioned in the introduction, different explanations have been proposed for the occurrence of multiple peaks in seasonal age spectra. Bönisch et al. (2009) refined the work by Andrews et al. (2001) and assumed that age spectra in the lowermost stratosphere (below about 380 K) result from the superposition of two single peak spectra, related to a fast (quasi-horizontal mixing and shallow circulation branch) and a slow pathway (deep circulation branch). They noted that this superposition does not necessarily cause a bimodal spectrum shape for sufficiently strong overlap of the individual spectra. Reithmeier et al. (2007) found several peaks in their age spectra, but only at polar latitudes. They explained these peaks as resulting from the





annual cycle in tropical upward mass flux into the stratosphere, maximizing during NH winter, and the seasonal variation in the polar vortex transport barrier, allowing horizontal transport to high latitudes only in spring and summer. This results in the superposition of single mode spectra from low latitudes once per year. Li et al. (2012a) also found multi-peak age spectra in their model simulation at polar latitudes and argued that they form mainly due to the fact that air masses leaving the boundary

layer during NH summer would have the highest probability to recirculate into the polar stratosphere.

As indicated in Fig. 6, we find multiple peaks as a generic feature throughout the lower stratosphere, although they are most pronounced at high latitudes. Closer inspection of the peaks shows that they correspond to NH winter pulse release times (cf. peaks in BIRs in Fig. 1a). To further highlight this point, Fig. 11a presents the distribution of the source times of the age spectrum peaks, hence the time when the air corresponding to the peaks has left the boundary source region at the Earth's

surface. The maximum frequency for NH winter source times clearly shows that air leaving the surface layer during winter is overrepresented compared to air leaving the surface during summer. Figure 11b further shows that also for the full age spectrum (not only the peaks) air leaving the boundary surface during NH winter is most likely. The link of the age spectrum peaks to an enhanced probability of stratospheric air to originate at the surface during NH winter compared to summer explains also the occurrence of the peaks at approximately the same transit time independent of latitude and level (Fig. 6), and is consistent with

the interpretation by Reithmeier et al. (2007).

One explanation for the enhanced occurrence of NH winter surface air in the spectrum is that air parcels preferentially enter the stratosphere during NH winter (cf. argument in Reithmeier et al., 2007). Another possible explanation is that vertical lofting in the tropical stratosphere only occurs efficiently during NH winter when tropical upwelling is strongest and the tropical stratosphere is well isolated. Some of this air then reaches the region of interest through a more or less direct pathway. Some of

it recirculates and reaches the region in following years. Pulses released during NH spring and summer, on the other hand, are efficiently dispersed meridionally before they reach the tropical pipe and therefore are less likely to undergo recirculation in the stratosphere. A sensitivity calculation with pulses released at the tropical tropopause (not shown) also results in age spectra with annually repeating peaks, however weaker, and therefore all transport processes described above are likely to play a role.

At first glance, the aging of the peaks by 1 month per month at a given location in the stratosphere (e.g. Fig. 4) would be

consistent with air simply staying at that location. This is possible in the extratropical middle and upper stratosphere during summer where the circulation essentially shuts down. But it is inconsistent with the aging of the peaks throughout the year - during winter the air is expected to sink, for example. Hence, the propagation of the peaks to older transit times (Fig. 4) results from the interplay of various processes.

Figure 6 shows that maximum tropical upward mass flux during NH winter causes a strong peak at short transit times in the

tropical lower stratosphere age spectrum. This tropical young air is transported rapidly by isentropic mixing and the shallow circulation branch to middle and high latitudes during spring and summer when the subtropical jet transport barrier is weak and the polar vortex transport barrier is absent. Therefore, the young air peak extends from the tropics to the summer pole (see Fig. 6a–b), particularly in the NH. A fraction of the tropical air, however, is transported through the deep circulation branch and arrives at higher latitudes during the following winters, when the deep circulation branch is most active. As this air corresponds

to the same winter source time as the air which was transported directly to higher latitudes at lower levels, it effectively creates





the age spectrum peak at older transit times and compensates the diluting effect of downwelling. Some of the extratropical air recirculates into the tropics and amplyfies the peaks in the spectrum tail. As the deep circulation branch is strongest during NH winter, the older spectrum peaks appear stronger in winter than summer (e.g., Fig. 6a–b). Mixing causes attenuation of the peaks.

Other modes of variability in transport, such as the QBO, may also modify the shape of the age spectrum and the occurrence of multiple peaks (see Fig. 9 and related discussion). The existence of a shallow and fast transport pathway further affects the shape of age spectra in the lowermost stratosphere. Transport by the deep circulation branch is likely responsible for the amplification of the older peaks at high latitudes (Fig. 6), but is not creating an additional peak, in agreement with Bönisch et al. (2009).

In contrast to Reithmeier et al. (2007) and Li et al. (2012a), the CLaMS age spectrum peaks are more pronounced, with multiple peaks occurring also in the subtropics and mid-latitudes, and even weakly so in the tropics (e.g., Fig. 6). Problems with representing the subtropical transport barrier in ECHAM4 (see Reithmeier et al., 2007) likely caused a masking of the multiple peaks at these latitudes in their study. Li et al. (2012a) used a climate model (GEOSCCM) that likely contained stronger numerical diffusion compared to the Lagrangian transport model used here. On the other hand, CLaMS is driven by
reanalysis winds, which are known to be overly dispersive (Schoeberl et al., 2003). A detailed understanding of the occurrence of multiple spectral peaks at lower latitudes requires further work.

## 6.2   Relation between age spectrum characteristics and mean age

Figure 12a–b shows the fresh air fraction with transit times shorter than six months ($F_6$) for NH winter and summer. The fraction of fresh air above the tropical tropopause is larger in NH winter than NH summer, consistent with maximum tropical
upwelling during NH winter. Although the strongest upwelling is slightly displaced into the respective summer hemisphere (e.g., Seviour et al., 2011; Abalos et al., 2015), the corresponding $F_6$ in this region is lower than in the winter hemisphere tropics and subtropics (Fig. 12a–b). This is a clear indication for strong horizontal mixing occurring in the summer hemisphere subtropics above the tropopause. Such mixing increases the amount of old air at low latitudes, consistent with an enhanced mass fraction of air older than two years in this region (Fig. 12c–d).

In the extratropics, the largest $F_6$ are found during summer in each hemisphere. In particular the NH lowest stratosphere is flushed with young air during summertime, increasing the $F_6$ there to about 20%. This flushing with young air has implications for the chemical composition of this region (e.g., Hegglin and Shepherd, 2007; Bönisch et al., 2009).

    Near the tropopause, mean age contours closely follow those of $F_6$ (Fig. 12a–b), as expected. The difference between mean age and $F_6$ increases with altitude and latitude. As mean age is strongly affected by the age spectrum's tail, its structure more
closely resembles that of the fraction of old air with transit time larger than two years (Fig. 12c–d). Especially the Southern polar vortex during June–August shows very old mean age and an almost vanishing fresh air fraction $F_6$ (Fig. 12b/d).

    Geographical distributions of the fresh air fraction and mean age at 400 K are shown for winter and summer in Fig. 13. Clearly, the largest $F_6$ occur during NH winter directly above the tropopause over the Western Pacific, the Indian Ocean, and Central Africa. During NH summer, the largest $F_6$ occur in a narrow band south of the equator and within the Asian





monsoon anticyclone (from the Tibetan plateau and North India to the Middle East). The enhanced young air mass fractions in the Asian monsoon likely have implications for pollution transport into the lower stratosphere (see also Randel et al., 2010), as the Asian monsoon system is close to the geographical source regions of highest anthropogenic pollution in India and China. Interestingly, both the fresh air fraction and mean age show a clear planetary wave signature at middle and high latitudes during

NH winter.

The geographical variability of the fresh air fraction at 400 K largely resembles the mean age distributions (Fig. 13c–d). Mean age therefore represents a good proxy for the spatial and temporal variability of the fraction of young air near the tropopause. With increasing altitude and latitude, however, mean age correlates more with the fraction of old air, which is more representative of the spectrum's tail, as noted above.

Figure 14 illustrates that the full age spectrum provides much more information than the mean age. The figure presents climatological wintertime age spectra along a given mean age contour of 2.5 years. Clearly, the spectrum shape significantly varies for the same mean, with a narrow spectrum at low latitudes and substantially broader spectra at higher latitudes. Estimating the full age spectrum and quantifying the amount of young air and its variability and changes is important for understanding the transport of short-lived chemical species and pollution into the stratosphere. This information requires evaluation of the full age

spectrum and cannot be estimated from the mean age alone. Knowledge of stratospheric age spectra is therefore advantageous for understanding lower stratosphere transport and composition.

### 6.3 Age spectrum approximation with 1D diffusion Green's function

For tracers with long enough chemical lifetimes stratospheric transport may be approximated by a one-dimensional flux-gradient relationship (e.g., Plumb and Ko, 1992). Based on this property, Hall and Plumb (1994) argued that the stratospheric

age spectrum may be well approximated by the Green's function for a one-dimensional diffusion process (denoted $\widetilde{G}^{\mathrm{diff}}$ in the following). Their derivation required the assumption of stationary flow (see e.g. Holzer and Hall, 2000, for a more general discussion). The 1D diffusion Green's function can be parametrised in terms of the mean age and spectrum width as (e.g., Bönisch et al., 2009)

$$\widetilde{G}^{\mathrm{diff}}(\tau) = \sqrt{\frac{\Gamma^3}{4\pi\Delta^2\tau^3}} \cdot \exp\left(-\frac{\Gamma(t-\Gamma)^2}{4\Delta^2 t}\right). \tag{7}$$

This 1D diffusion Green's function is generally used to approximate the age spectrum shape for deducing age of air information from observed tracers (e.g., Schoeberl et al., 2005).

A comparison of a single annual mean age spectrum in the NH lower stratosphere to this idealized age spectrum was presented earlier in Fig. 3. Here, the idealized stationary spectrum (Eq. 7) was fitted to the age spectra from CLaMS by using a least squares method (starting with the mean age and width of the simulation). The comparison shows that the annual mean

age spectrum is remarkably well described by the idealized age spectrum for stationary flow. Seasonal age spectra, however, are not well characterized by the idealized functional behaviour of Eq. (7), mainly due to the existence of multiple peaks.

To put this result into a global context, we apply the same fitting procedure globally and present the root mean square deviations between the climatological annual mean CLaMS spectra and the idealized fits, i.e. $\mathrm{RMSE} = \sqrt{\frac{1}{N}\sum_{k=1}^{N}\frac{[G(\tau_k)-G^{\mathrm{fit}}(\tau_k)]^2}{G(\tau_k)}}$





(Fig. 15a). Smallest deviations and best fits emerge in the extratropics above about 500 K. At latitudes equatorward of about 30°N/S deviations increase and the goodness of the fit degrades. For the annual mean spectra, deviations are generally smaller and therefore the fit is more accurate than for the individual seasons (not shown).

Examples for the age spectrum fit are presented in Fig. 15b/c. In the extratropics above about 500 K, the annual mean age spectrum is very well approximated by the 1D diffusion Green's function (not shown). In the tropics (here at 600 K), on the other hand, the idealized fit function decays too strongly (Fig. 15b). The longer tail in the CLaMS age spectrum in this case indicates the impact of recirculation on composition in the tropical pipe. This process can evidently not be described by simple one-dimensional diffusion.

During single seasons the goodness of the fit is generally less than for the annual mean (larger RMSE). In particular the NH lower stratosphere below about 500 K and the winter polar vortex regions show large deviations, presumably related to the strong seasonality of transport in these regions (not shown). The seasonality of transport causes multiple peaks in the age spectrum, as discussed in Sect. 6.1. Figure 15c shows the age spectrum in the NH lower stratosphere at 400 K and 60°N during spring (March–May), as an example of distinct annual peaks (same spectrum as the green line in Fig. 3). Fitting a single 1D diffusion Green's function $\widetilde{G}^{\text{diff}}$ to this distribution is clearly inapt. A fit to a superposition of three such functions, hence to

$$G^{\text{fit}} = \sum_{i=1}^{N} \alpha^i \, \widetilde{G}_i^{\text{diff}}, \tag{8}$$

with $N = 3$, however, describes the age spectrum very well (red curve in Fig. 15c). This fitting procedure is similar to the methodology introduced by Bönisch et al. (2009), motivated by Andrews et al. (2001). However, the interpretation of the multiple spectral peaks here is different. Bönisch et al. (2009) related the occurrence of two spectral peaks in the extratropical lowermost stratosphere to the existence of distinct transport pathways (shallow/fast vs. deep/slow circulation branches). Our analysis shows that multiple peaks are a generic characteristic of age spectra in the lower stratosphere and are caused by seasonality in transport (see Sect. 3 and Sect. 6.1). Hence, an accurate fit of age spectra in the lower stratosphere requires the superposition of a number of 1D diffusion Green's functions, with this number equal to the number of distinct peaks in the spectrum.

## 7 Conclusions

We presented seasonally and inter-annually varying age spectra in the lower stratosphere calculated with the Lagrangian transport model CLaMS driven by ERA-Interim winds for the period 1979-2013. Our approach is based on the boundary impulse response (BIR) method (e.g., Haine et al., 2008; Li et al., 2012a), generalized to transient simulations using quasi-observational winds, and is therefore more appropriately termed the Boundary Impulse (time-)Evolving Response (BIER) method.

Seasonal age spectra in the lower stratosphere show large deviations from an idealized stationary unimodal shape. Multiple peaks emerge in the age spectra throughout the stratosphere (strongest at high latitudes), caused by the interplay of seasonally varying tropical upwelling and stratospheric transport barriers. These multiple peaks are largely related to the fact that air entering the stratosphere during NH winter makes up the biggest fraction throughout the stratosphere. While the annual mean





spectrum is in large parts well described by a Green's function representative of idealized stationary transport, seasonal age spectra can only be well approximated by a superposition of such functions.

In addition to seasonality, inter-annual variations in transport (e.g., QBO, ENSO) cause significant age spectrum modulations. Stronger upwelling during both easterly QBO and El Niño phases increases the fraction of young air in the spectrum.

We found that the QBO phase may determine the modal age (age spectrum maximum) for up to 3 years across a wide range of latitudes.

Interpretation of the age spectrum in terms of residual circulation and mixing is not straightforward, with different effects dominating in different atmospheric regions. We found residual circulation trajectories, which are much more easily obtained than age spectra or even mean age, to represent a good approximation of the dominant pathway in the deep tropics and in the

winter extratropics above about $500\,\mathrm{K}$, as given by the modal age in these regions. In contrast, eddy mixing strongly modifies the modal age in summer, particularly in the lowermost stratosphere.

Mean age appears to be a reliable proxy for spatial variations in the age spectrum just above the tropopause, but less so throughout most of the stratosphere. Knowledge of the exact fractions of air masses with certain transit times, as included in the full age spectrum, is therefore highly beneficial for a more detailed understanding of stratospheric chemistry and compo-

sition. Including age spectrum diagnostics in state-of-the-art atmospheric climate and transport models would help to avoid ambiguities in model inter-comparisons of stratospheric transport.

**Appendix A:  Correction for finite age spectrum tail**

The tail of the age spectrum $G(\tau)$ in the lower stratosphere generally decreases exponentially at transit times larger than about 4-5 years (e.g, Fig. 1c), as noted by several authors (e.g., Reithmeier et al., 2007; Diallo et al., 2012; Li et al., 2012a). At large

transit times $\tau > \tau^*$, larger than some threshold $\tau^*$, the spectrum tail may therefore be approximated by an exponential function to define a *corrected age spectrum* by

$$G_{\mathrm{corr}}(\tau) = \begin{cases} G(\tau) & \text{for } \tau \leq \tau^* \\ G(\tau^*)\,e^{-\frac{\tau}{\xi}} & \text{for } \tau > \tau^* \, . \end{cases} \tag{A1}$$

As our multi-pulse set-up in CLaMS with 60 pulse tracers every two months (see Sect. 2.2) allows calculating the age spectrum for transit times up to ten years, we chose $\tau^* = 10$ years and determine the decay time scale $\xi$ by fitting the exponential function

for transit times $5\,\mathrm{yr} < \tau < 10\,\mathrm{yr}$.

Integration of the corrected age spectrum (Eq. A1) over transit time (from zero to infinity) yields the corrected norm and the first moment of the age spectrum (the corrected mean age) (see also Diallo et al., 2012, Eq. 2)

$$N_{\mathrm{corr}} = N + G(\tau^*)\,\xi \, , \tag{A2}$$

$$\Gamma_{\mathrm{corr}} = \Gamma^* + G(\tau^*)\,\xi\,(t^* + \xi) \, . \tag{A3}$$

Figure 16 shows the reference (uncorrected) mean age from the CLaMS climatological age spectrum for winter and summer and the difference to the corrected mean age (corrected minus reference). As expected, the correction of the spectrum tail up to



infinite transit times causes older mean age globally. The differences increase with increasing latitude and altitude, remaining below about 6 months throughout large parts of the lower stratosphere, but increasing to about 1 year inside the polar vortex (particularly in the SH).

*Acknowledgements.* We thank John Bergman, Harald Bönisch, Paul Konopka and Bernard Legras for helpful discussions and suggestions.
5   Thanks also to Nicole Thomas for programming support and to the ECMWF for providing reanalysis data. This work was funded by the German Federal Ministry of Education and Research (BMBF) within the "ROMIC" programme under project 01LG1222A. Felix Ploeger was funded by a HGF postdoc grant.



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



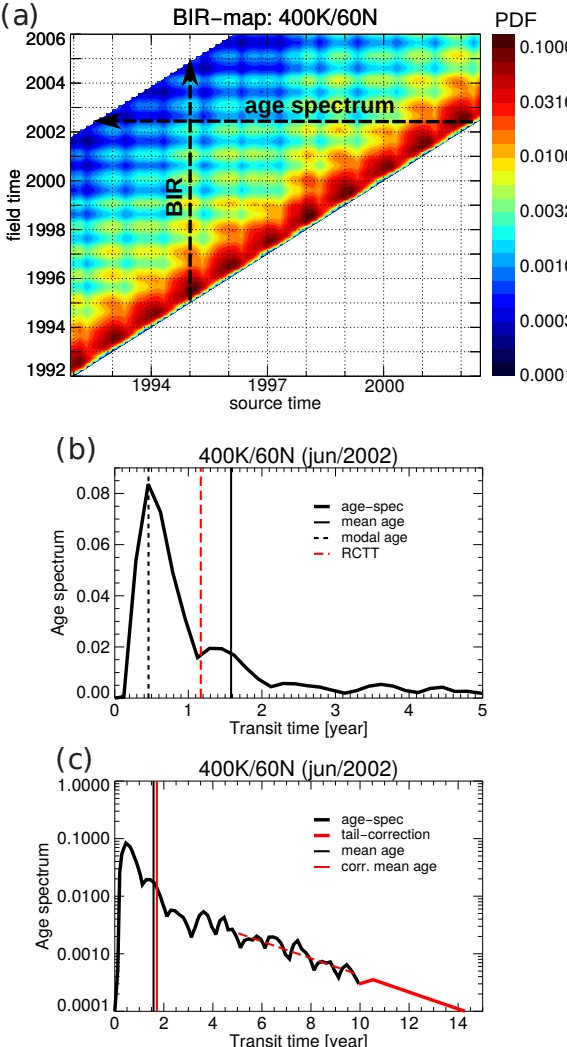

**Figure 1.** (a) BIR map from CLaMS pulse tracers at 400 K and 60°N, plotted only for part of the full simulation period. Arrows highlight vertical and horizontal cuts through the BIR map representing BIR (vertical) and age spectrum (horizontal), respectively (cf. Haine et al., 2008). (b) The age spectrum for June 2002, corresponding to the horizontal cut in (a). Vertical lines show mean age (solid), modal age (black dashed) and the residual circulation transit time (red dashed, see text). (c) Same as (b), but including a correction for the finite spectrum tail using an exponential fit (see text). The red vertical line shows mean age for the tail-corrected spectrum, the black line for the uncorrected case (Note the larger transit time range and logarithmic y-axis).

Wolter, K. and Timlin, M.: Monitoring ENSO in COADS with a seasonally adjusted principal component index, in: Proc. of the 17th Climate Diagnostics Workshop, Norman, OK, NOAA/NMC/CAC, NSSL, Oklahoma Clim. Survey, CIMMS and the School of Meteor., Univ. of Oklahoma, pp. 52–57, 1993.

Wolter, K. and Timlin, M.: Measuring the strength of ENSO events - how does 1997/98 rank?, Weather, 53, 315–324, 1998.





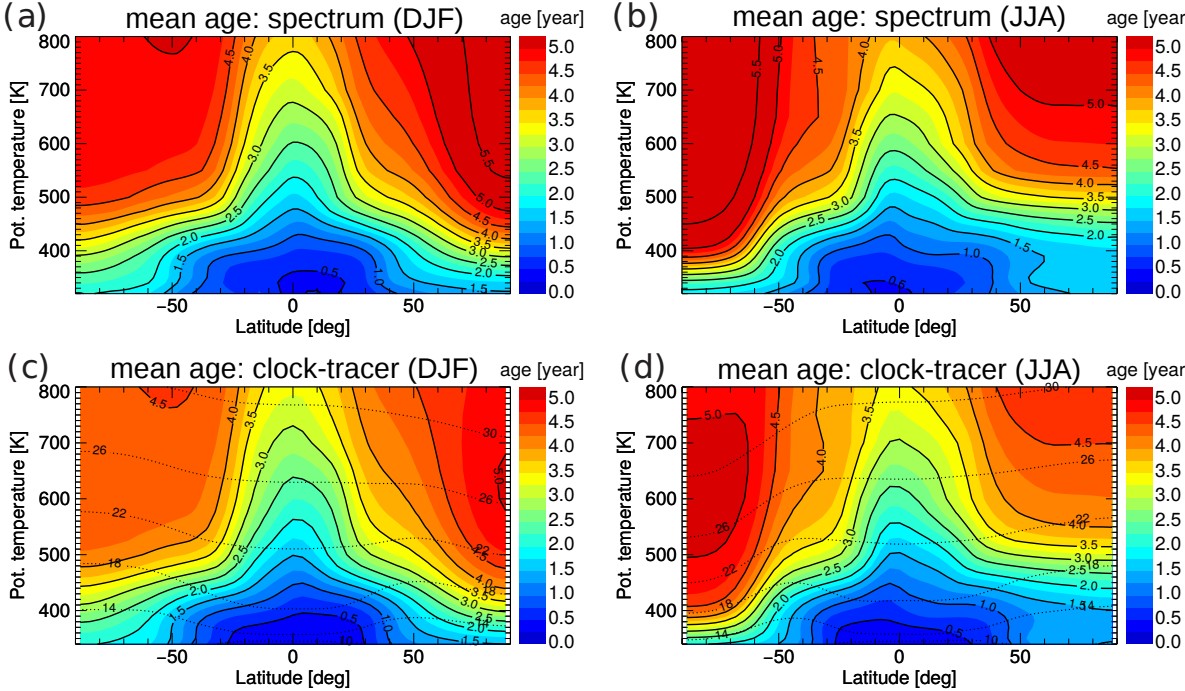

**Figure 2.** Mean age from the 1989–2013 CLaMS age spectrum climatology for (a) December–February and (b) June–August. Corresponding mean ages calculated from the model's 'clock-tracer' are shown in (c) and (d) for comparison. Black dotted lines in (c/d) show altitude levels in km for reference.

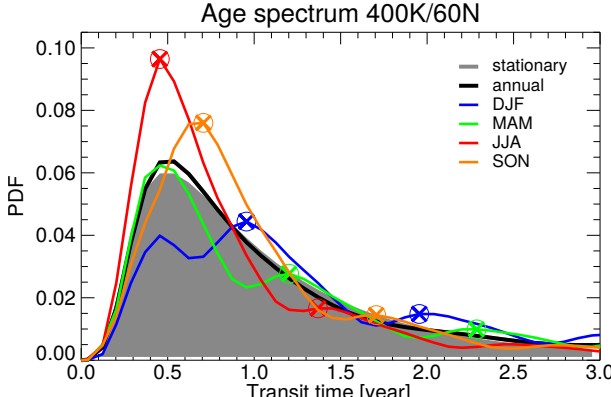

**Figure 3.** Seasonality of the climatological age spectrum at 400 K in NH mid-latitudes (60°N). The annual mean spectrum is shown as a black thick line, the idealized stationary shape (see text) as grey shading, and different seasons as coloured lines. Symbols illustrate the evolution of the peak (see text for details).



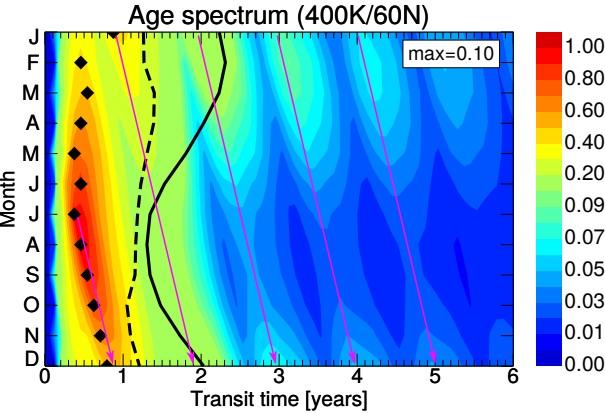

**Figure 4.** Seasonal cycle of the climatological age spectrum at 400 K in the NH extratropics (60°N). Mean age is shown as black thick line, modal age as black symbols, and RCTT as black dashed line. The pink arrows illustrate the time evolution of the spectrum peaks. (Note the non-linear colour scale).

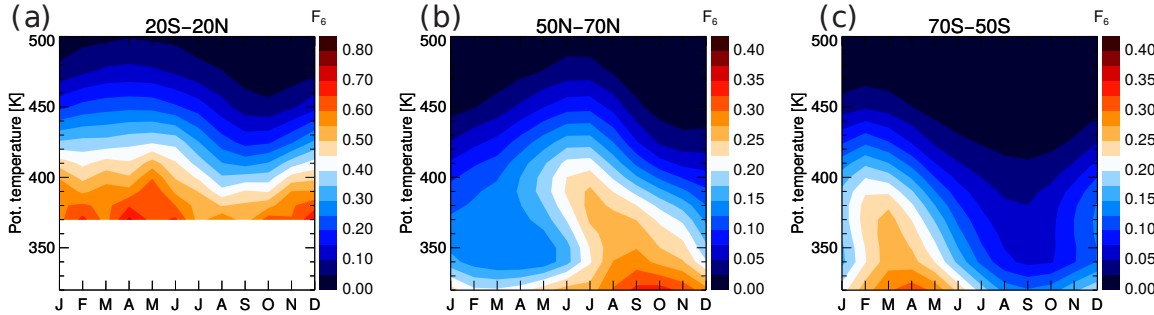

**Figure 5.** Seasonality of fresh air fraction (transit time below six months) in (a) the tropics (20°S–20°N), (b) the SH extratropics (70°–50°S), and (c) NH extratropics (50°–70°N). (In the tropics (a), the region below 370 K is left white, as the pulse separation of 2 months lies in the range of the mean transit time and the calculation of the fresh air fraction becomes unreliable.)





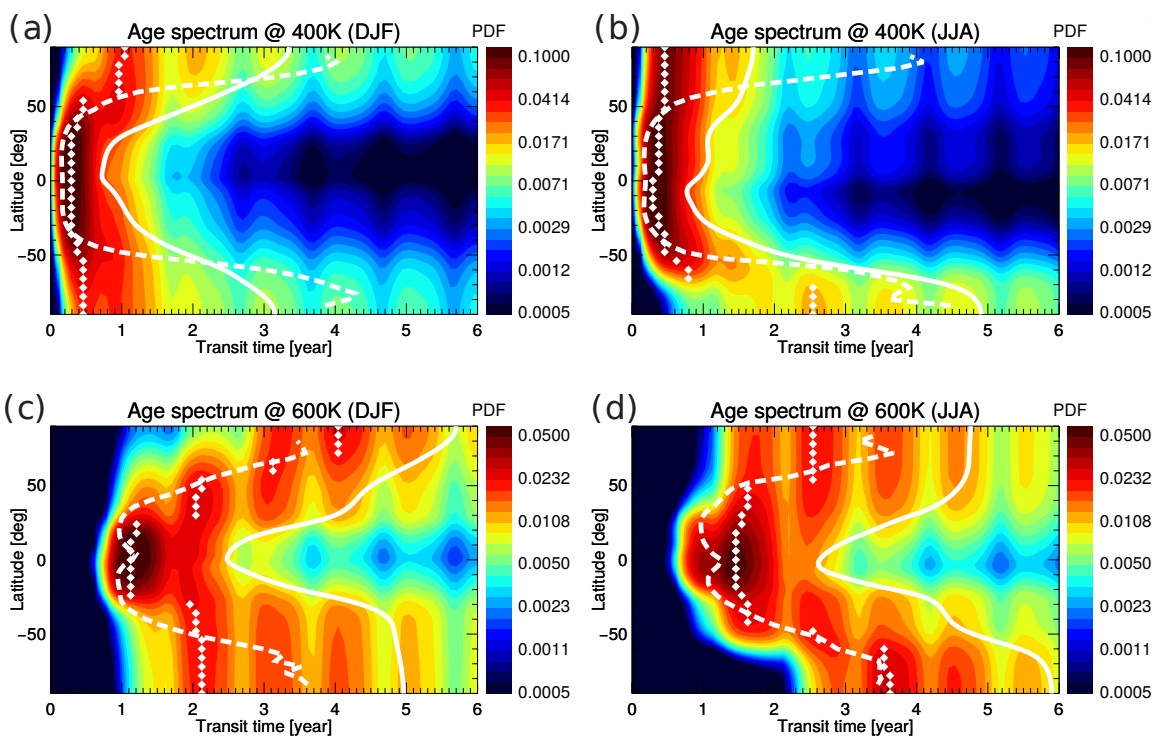

**Figure 6.** Age spectra at 400 K for (a) December–February and (b) June–August. (c) and (d) ahow the same but at 600 K. White lines show mean age (solid) and the residual circulation transit time (dashed). White diamonds show modal age.



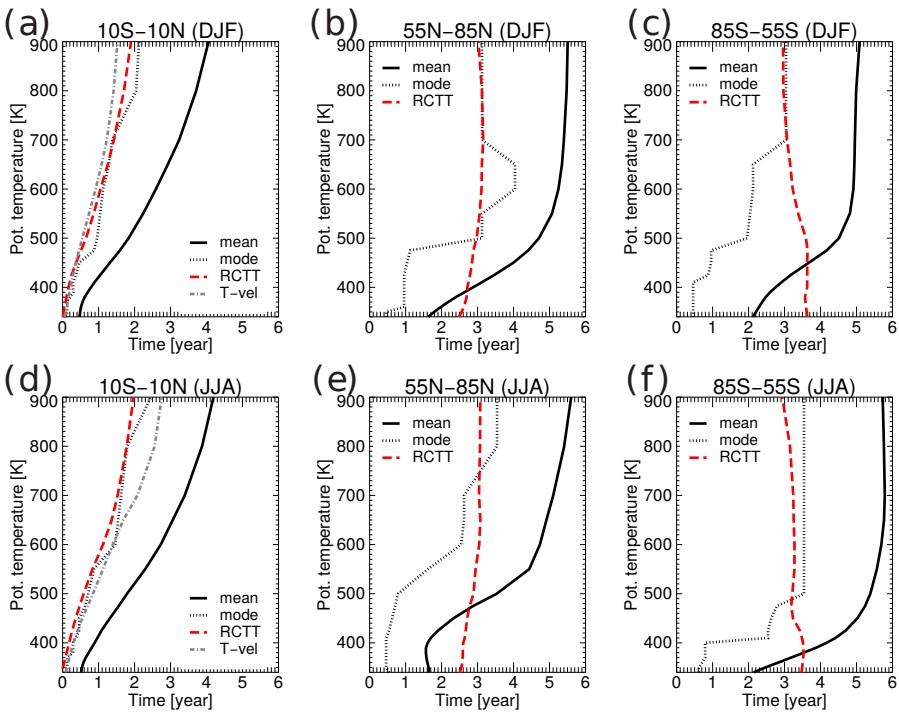

**Figure 7.** Profiles of mean age, modal age and RCTT for DJF (top) and JJA (bottom) at the equator, in the NH between 55°N–85°N, and in the SH between 55°S–85°S (from left to right). The grey dashed line in (a) and (d) shows the mean tropical upwelling time scale estimated from the zonal mean vertical velocity.





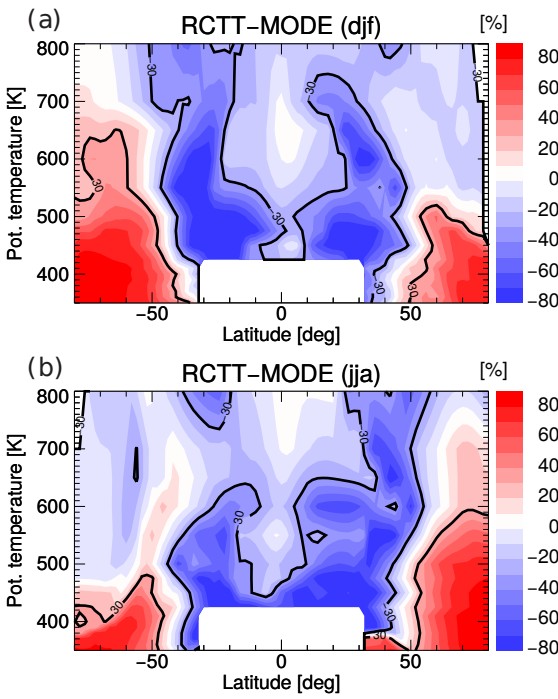

**Figure 8.** Difference between RCTT and modal age (RCTT − mode)/RCTT in percent for (a) December–February, and (b) June–August. The black line highlights the 30% contour. The tropical lowest stratosphere (below 420 K) is left white as relative differences become arbitrarily large there due to small transit times.



**Figure 9.** Age spectrum time series at 600 K and 10°S–10°N (top), 40°N–60°N (middle), and 60°S–40°S (bottom). Shown are the spectra (left) and their deseasonalized anomalies (right). Black lines show mean age, symbols modal age, and red lines show RCTT (full values on the left and their respective deseasonalized anomalies on the right). Red horizontal bars highlight easterly QBO phases.



**Figure 10.** Same as Fig. 9 but at 400 K. Red horizontal bars highlight El Niño periods (Multivariate ENSO Index above 0.5, see Wolter and Timlin, 1993, 1998).





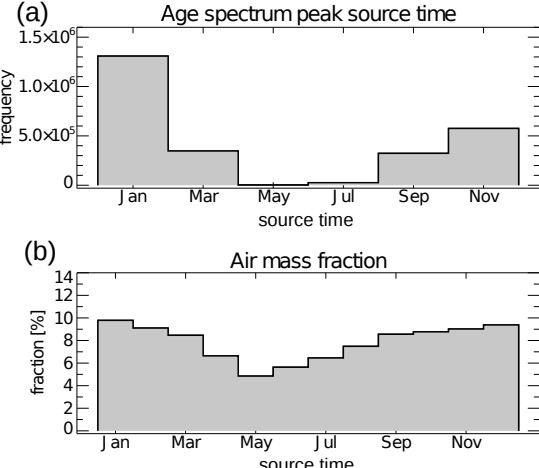

**Figure 11.** (a) Distribution of time of last contact with the boundary surface (source time) for age spectrum peak air. (b) Seasonality of time of last contact with the boundary surface for all stratospheric air masses. The distributions have been calculated from all CLaMS age spectra in the 380–1000 K potential temperature layer during the 1989–2013 period.

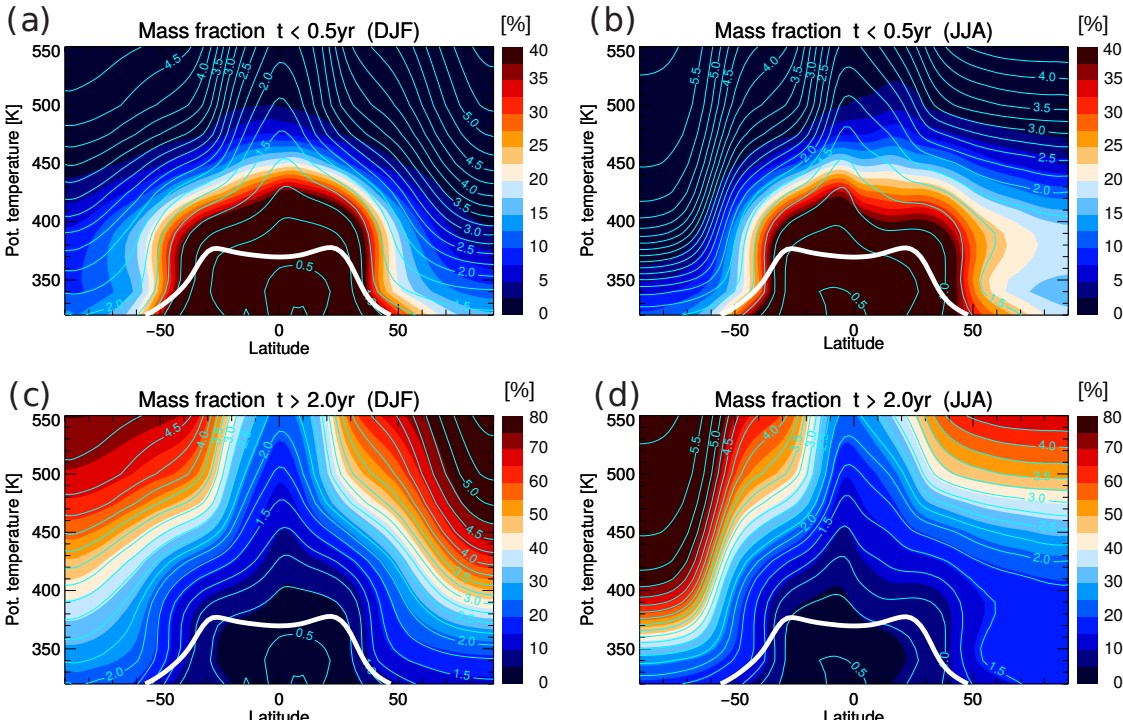

**Figure 12.** Fresh air fraction $F_6$ (transit time below 6 months) during (a) December–February, and (b) June–August. (c) and (d) show the fraction of old air (transit time above 2 years). The white contour shows the thermal tropopause, cyan contours show mean age.



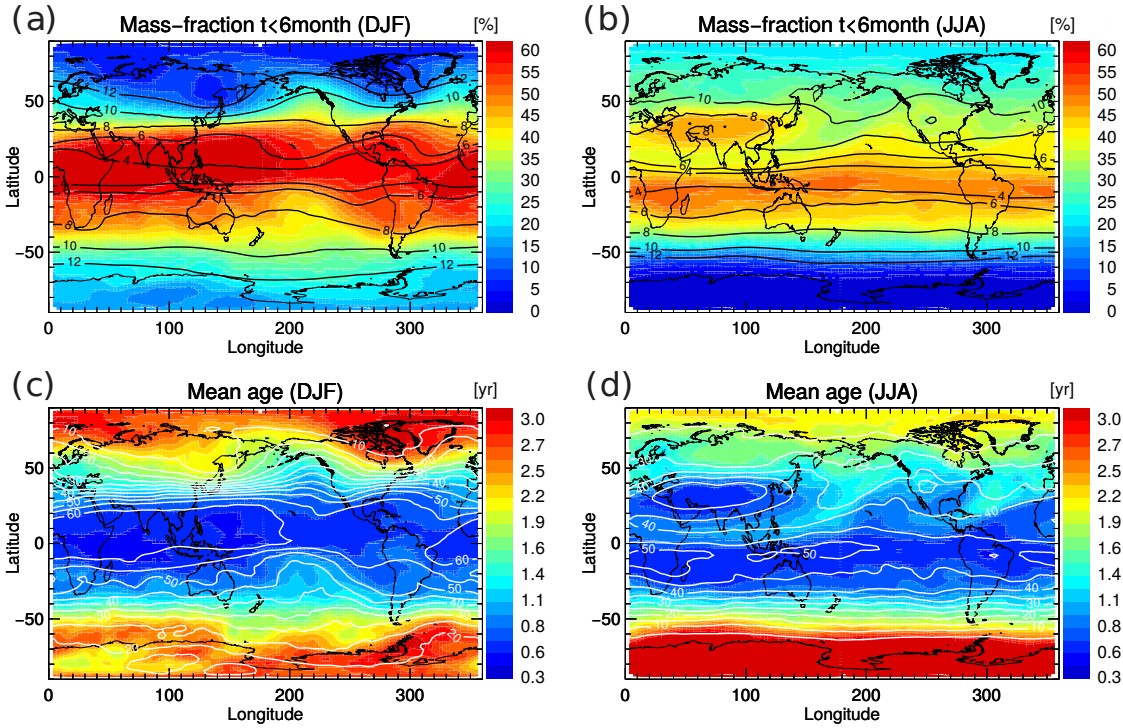

**Figure 13.** Fresh air fraction $F_6$ (transit time below 6 months) at 400 K during (a) December–February, and (b) June–August. Black contours show potential vorticity from ERA-Interim ($\pm4$, $\pm6$, $\pm8$, $\pm10$, $\pm12$ PVU). (c) Mean age at 400 K during December–February, and (b) June–August (white contours illustrate the fresh air fractions from a/b).

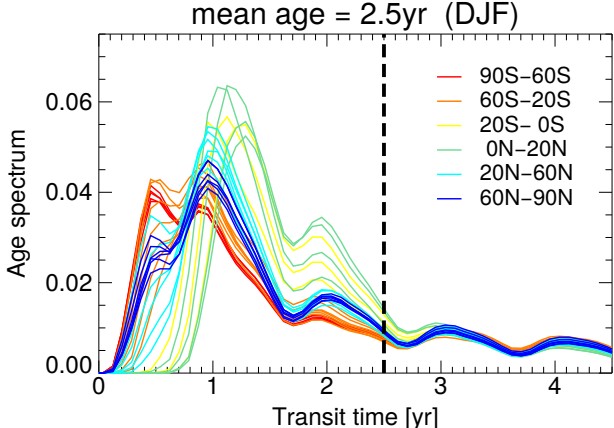

**Figure 14.** Climatological age spectra for the same mean age of 2.5 years but at different latitude ranges (marked by colours), for December–February. The black dashed line highlights the mean age of 2.5 years.





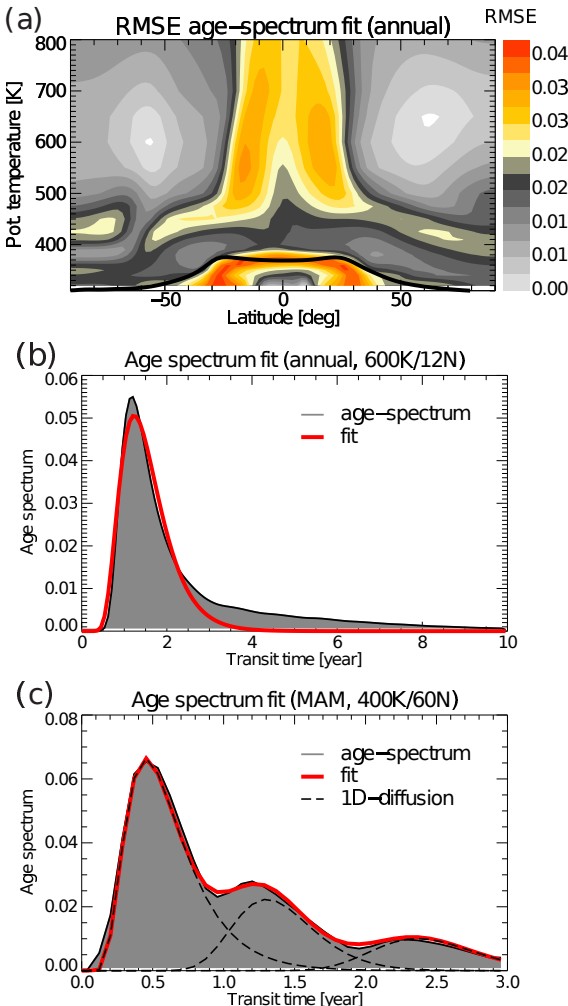

**Figure 15.** (a) Root mean square deviation (RMSE) of the fit of climatological (1989–2013) annual mean CLaMS age spectra with the idealized stationary age spectrum of Hall and Plumb (1994) (1d diffusion Green's function). The black line shows the thermal tropopause (from ERA-Interim). (b) Climatological annual mean age spectrum at 600 K in the tropics (12°N) from the CLaMS simulation, and the corresponding fit using the 1d diffusion Green's function (red). (c) Climatological spring (March–May) age spectrum in the NH lower stratosphere (400 K), and a fit using a superposition of three idealized stationary spectra (red).





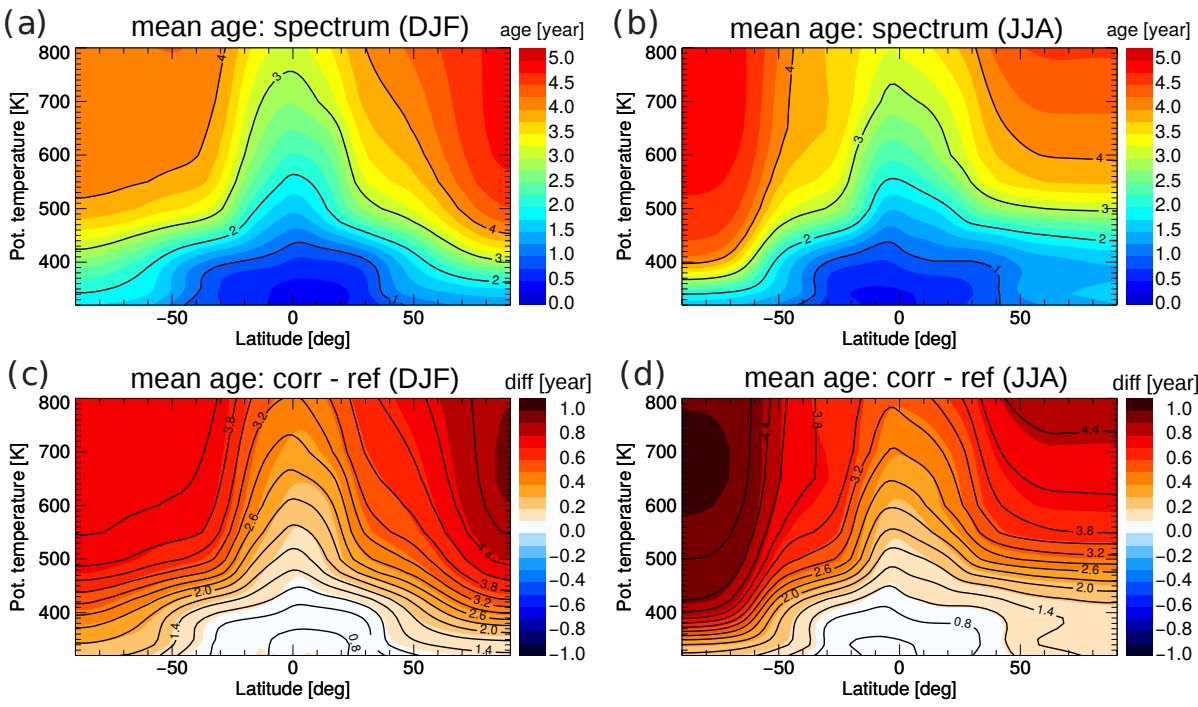

**Figure 16.** Climatological mean age from CLaMS age spectrum for (a) December–February and (b) June–August, without applying the tail-correction (see text). (c) and (d) show the difference of age spectrum mean ages with correcting for the finite spectrum tail to this reference. Black lines highlight particular mean age contours.