# Peer review of "Seasonal and inter-annual variability of lower stratospheric age of air spectra"

_Atmospheric Chemistry and Physics, 2016_

## Referee Comment (RC1) · Anonymous Referee #1 · 16 Mar 2016

General comments:

This paper is very well written, enlightening, and concerns an important and timely aspect of atmospheric transport. My primary criticism is that the discussion is too terse in places. I suggest the addition of strategically placed clarifying phrases will help the reader understand the logic of the arguments. In the specific comments, I have noted a few examples where I had a particularly difficult time following the logic. In addition, the discussion of the relationship between ENSO and inter-annual variations of the age spectrum (Figure 10 and the last paragraph of Sec. 5) is not convincing. It should either be improved with some quantitative measures of the information content of that relationship or removed.

Specific Comments:

[Figure]

Page 2, line 17: The discussion in this paragraph could greatly enhanced if you briefly explained (1) the motivation for representing transport as a diffusion process and that for representing the age spectrum by a Green's function, (2) the assumption implicit in those representations, and (3) why the assumptions needed for such a representation are not met.

P. 2, L. 34: Changing 'using multiple pulses' to 'using multiple tracer pulses' will help clarify what you are doing.

P. 6 L. 16-19: Perhaps I am confused here. Shouldn't the mean age calculated from the age spectrum be identical to the value calculated from a perfectly linear clock tracer? Why is there a (small) discrepancy between the two calculations? Is it because, for ages approaching 10 years, the clock tracer does not capture the full range of source concentrations (i.e., the clock does not start early enough)?

P. 6 L. 26-27: It took me a while to decipher the meaning of the phrase 'coinciding with the youngest peak from spring to fall and the second youngest peak in winter'. Perhaps it would help to be more explicit, for example, 'the modal age as determined by the apex of the largest peak coincides with the youngest peak during spring, summer and fall and with the second peak during winter.'

P. 7 L. 9-12: The description of the propagation of the peak and the argument that the peaks are due to the fact that the most efficient transport from the boundary layer to the stratosphere is too terse. Perhaps another sentence or two will clarify the argument. For example, I can understand why efficient transport during winter leads to a peak in the summertime age spectrum at 6 months. However, I don't understand why the subsequent wintertime peaks are also linked to transport efficiency during winter. It seems to me that once the air is in the stratosphere (i.e., after 6 months) then an additional boundary layer to stratosphere transport boost the following winter is irrelevant; unless an important fraction of the air released during a specified winter remains in the troposphere until the following winter – when the troposphere-to-stratosphere 'transport

window' re-opens.

P. 7 L. 24-25: Vertical velocity near the tropopause is also slower during summer than during winter (W. Randel and co-authors have written papers on this) and could be important.

P. 10 L. 8-20: The connection between ENSO and inter-annual variations of the age spectra is, perhaps, believable, but is not convincingly demonstrated in Fig. 10. This could be due to the fact that, while equatorial convection patterns shift substantially as sea surface temperature patterns shift, variations of the average strength of tropical convection have relatively weak connections to ENSO. It seems the authors are trying to see make too much of patterns that appear in Fig. 10 – a problem that can arise when analysis rely too heavily on a qualitative comparison and the ability of the human eye to recognize patterns in a chaotic system (whether or not the patterns are meaningful). At the very least, Fig. 10 should be changed to make it easier to discern how well the age spectra are related to ENSO. It would be better to make the analysis more quantitative. For example, how much to the inter-annual variance of the age spectra is explained by a lag-relationship with ENSO variability? It might be best to simply remove Fig. 10 and the last paragraph of Sect. 5. Fig. 11: Did you create this plot for the global release experiment? (as opposed to 15S-15N) Does it look the same for both experiments? If so, or even if not, that is an important comparison to make.

P. 11 L. 11-12: Does the distribution in Fig. 11b mean that the particle pulses do not always have the same mass? That the rate of particle release varies? If so, how is this justified physically? Regardless, please explain this figure a bit more.

Sec. 6.1, last 3 paragraphs: This discussion could use elaboration. Please be sure that, each time the effect of some phenomenon on peaks is mentioned, the explanation for how (or why) the effect is carried out is clear.

P. 12 L. 26-7: Regarding 'This flushing . . . has implications for the chemical composition . . .'. Can you give an example?

Technical details:

P. 2 L. 16: 'Many studies of stratospheric age . . .'

P. 3 L. 2: Change 'spectra from seasonal . . .' to 'specta on seasonal . . .'

P. 8 L. 19: 'plays a dominant role' is an over-statement without further analysis. Change 'dominant' to 'important'. Better yet, use 'plays a more important role during this season than during NH winter'.

P. 8 L. 21: Change 'The isolation of tropical air through the subtropical transport barriers' to 'imposed by the subtropical transport barriers' (or some analogous change) to avoid the contradictory imagery invoked by the words 'isolation' and 'through'.

P. 9 L. 3: Change 'transit times above' to 'transit times longer than'.

---

## Referee Comment (RC2) · E. Ray (Referee) · 17 Mar 2016

This paper uses trajectories from the CLaMS transport model driven by ERA-Interim meteorology to analyze stratospheric age of air spectra. This work provides unique and revealing insights into the stratospheric transport on various time scales and latitude regions. The authors have recently done nice work on explaining various aspects of the stratospheric mean age of air and how the residual mean circulation and isentropic mixing contribute to the observed mean age distributions. But as they mention here, the age spectra provide another level of information, in particular how the transport variability imprints on the age spectra for years afterward.

The analysis is excellent and I highly recommend publication with consideration of the minor comments below. The paper is a bit long, 16 figures is quite a lot but I don't

have any specific suggestions on what could be left out. Perhaps with fewer figures a bit more time could be spent explaining some of the new and unique features of the remaining plots.

Minor comments:

Page 2, line 14: awkward sentence, change to something like "comes with the benefit of allowing one to separate..."

Page 2, line 35: "...and allows one to calculate the..."

Page 2: Just a comment that in my Ray et al. [2014] paper figure 4 shows age spectra from the TLP model with multiple peaks with clear seasonal and QBO influence. We didn't explain what caused the peaks beyond the known variability in the MERRA transport input to the model but thought you might want to include a mention of this paper here.

Page 7, line 13: "...allows one to quantify air..."

Page 8, lines 13-15: The younger tropical spectrum peak in DJF in Figure 6 is hard to see. I'll take your word for it but this figure doesn't seem to back up that statement unless I'm missing something.

Page 10, lines 1-8: Figure 9 is really nice and has so many features it takes some time to appreciate them all and what they mean. The propagation of the signals to older parts of the age spectra with time is reminiscent of the tropical tape recorder signal but has a different physical meaning. It's actually difficult to interpret physically what's going on with these signals because it's in a different phase space than we're used to thinking about. For instance, in Figure 9c-f there are anomalous peaks and troughs in the spectra that appear to propagate from the time when an event like a QBO easterly phase occurred at ages from 1-2 years for the following 4 years out to ages of 6 years. The actual air masses that were influenced by the QBO transport anomaly move around the stratosphere and many of them actually leave the stratosphere over

the following 4 years and yet at this theta level and latitude range there is a signal that remains. It's as though the air influenced by the particular QBO event circulates around and enough of it comes back through this theta and latitude region to maintain an anomalous signal. That's actually remarkable! It might be worth spending a bit more time explaining the physical meaning of this plot and the features since I don't think it's obvious and I'm not even sure I'm getting the full picture.

Figure 5: in the caption the hemispheres are switched for (b) and (c).

Figure 6: in the caption should be "show"

Figure 7: In the interest of shortening the paper could be one to consider removing.

Figure 9: Really interesting as mentioned above and a lot going on here. My suggestion to be able to see the features more clearly on b,d and f primarily is to separate out the delta age, mode and RCTT that are on the right axis into their own plots. It's hard to see their oscillation around the zero line as it is and it obscures somewhat the propagation of the pdf anomalies.

Figure 14: Is each of the 3 lines of each color an individual month within DJF?

---

## Referee Comment (RC3) · Anonymous Referee #3 · 19 Mar 2016

This manuscript presents calculations of the seasonal and inter annual variations in the stratospheric age spectrum obtained from the CLaMS model driven by ERA-Interim winds. The manuscript contains some interesting and new results that certainly warrant publication. However, the manuscript is not very well focused, does a poor job of discussing previous studies, and some of the major conclusions (as stated in abstract or conclusions) are not new results. I think the manuscript needs to be revised to be more focused on the new aspects of their calculations (seasonal and inter annual variability, including QBO) and to put these in context of previous studies.

MAJOR COMMENTS

1. The manuscript is not very focused, and the new results are not clearly presented. This lack of focus can be seen from the title, which is very vague (and doesn't match

much of the manuscript). In my opinion the relatively new and important aspect of the study is looking at seasonality and inter annual variability of the age spectrum, which has only been done in a few previous studies, and none of the previous studies have included the QBO (as the authors highlight in the Introduction). However there is actually relatively little discussion of these aspects, and there is just as much (or even more) discussion is on the age spectrum - mean age relationship, the approximation of G by inverse Gaussian, and comparisons of residual circulation with modal age. All of these latter issues could be examined using steady flows, and much of what is discussed is already known. My recommendation is to focus on the seasonal and inter annual variations (and QBO) and to minimize the discussion of the other issues.

2. While there are references to previous studies that examined similar aspects of the age spectrum in the Introduction there is very little discussion of these when interpreting the calculations presented here. Because of this it is not clear to me how much of the results are new and how much are just reproducing earlier results with Lagrangian model driven by reanalyses.

The clearest examples of lack of discussion of previous studies are Sections 3 (seasonality) and 5 (inter annual variability) where there is not a single mention of the Li et al. 2012a,b studies which examined exactly these issues. How do the results presented compare with these previous studies? What is new in what is presented (other than a slightly different approach)?

3. Some of the conclusions (as stated in the abstract) are well known facts. This paper may be providing more support, but as written it appears these are new results.

One example is the statement in the abstract that "Interpretation of the age spectrum in terms of transport contributions due to the residual circulation and mixing is generally not straightforward." This is well known and not sure this counts as a significant enough statement for an abstract.

Another example are the statements towards the end of both the abstract and conclu-

sions regarding benefits of age spectrum calculations and need for inclusion in model inter comparisons. Again not new, and in fact age spectrum calculations were actually included in model inter-comparisons in late 1990s (Hall et al. 1999). Is this really the major take home message from this manuscript (ending abstract and conclusions with this gives this impression)?

4. There are multiple places where I think previous papers or the current understanding are misrepresented. Examples:

Pg 1, Line 24: As written it suggests that the Hall and Plumb 1994 and Waugh and Hall 2002 papers did not appreciate that there was a range of pathways and an age spectrum, whereas the opposite is true and Hall and Plumb focused on this fact.

Pg 2, Line 6: I don't think it is correct to indicate that the apparent disagreement between observed and models changes in transport is due to age spectrum versus mean age differences. This disagreement occurs if you compare mean age from observations with mean age from models, so not a residual circulation vrs mean age issue.

Pg 2, line 28: Is it a common view that modal age can be related to residual mean mass circulation? Maybe in the tropical lower stratosphere, but I don't it is common to think such a relationship extends beyond this region.

Pg 13, line 19: It is not correct to say that "Hall and Plumb (1994) argued that the stratospheric age spectrum may be well approximated by the Green's function for a one-dimensional diffusion process". They used the 1 diffusion model to illustrate aspects of the age spectrum not to model the actual stratospheric age spectrum (they are explicit about this: "our goal at this point is not to quantify stratospheric transport, but rather to illustrate the points of the previous discussion").

Pg 14, line 7: "This process can evidently not be described by simple one-dimensional diffusion." I think this well known, not only from looking at G(t) from other three-dimensional models, but also from the tropical leaky pipe model where expressions

for the age spectrum have been derived (Hall 2000) (see also Waugh and Hall (2002) review).

MINOR COMMENTS

Pg 6, line 15-19: How long was the clock tracer run for; is it in (quasi-)steady state? It is stated that the agree is good but in polar regions the clock tracer is younger by over 0.5 yrs. Note, the paper of Hall and Haine (2002) might be relevant here. They derive the relationship between "ideal age" (which is an alternate clock tracer) and "mean of age spectrum". In their example the clock tracer converges quicker than the mean of the age spectrum, which appears opposite to your result. However, this may because you have run calculations for different length.

Eq 7: Why reference Bonisch et al 2009? As you have just mentioned the G(t) for 1D advection diffusion was used in Hall and Plumb, and this same expression was given in the Waugh and Hall (2002) review paper. Furthermore, this is not a parameterization, this is the exact G(t) just expressed in terms of the mean and width (rather than flow velocity and diffusion).

Section 5: Is it possible to make composites on east and west phase of the QBO?

Pg 11, line 16 -: Are the results in Orbe et al. (2014) relevant for this discussion?

REFERENCES

Hall, T. M., Path histories and timescales in stratospheric transport: Analysis of an idealized model, J. Geophys. Res., 105, 22,811–22,823, 2000.

Hall, T. M., and T. W. N. Haine, A note on ocean transport diagnostics: Ideal age and the age spectrum, J. Phys. Ocean- ogr., 32, 1987–1991, 2002.

Orbe et al. Seasonal ventilation of the stratosphere: Robust diagnostics from one-way flux distribution, JGR, 119, 2014.

---

## Author Comment (AC1) · 31 May 2016

We thank all three reviewers for their careful considerations of the manuscript and their well thought-out comments. These certainly helped to significantly improve the paper. In the following, we address all comments and questions raised (Reviewer's comments in italics). Text changes in the manuscript are highlighted in color (except minor wording changes). The main concerns of the reviewers were:
(i) 'The manuscript is not very focused.' (Reviewer 3)
(ii) 'The paper is a bit long.' (Reviewer 2)
(iii) 'The discussion is too terse in places.' (Reviewer 1)
(iv) 'The manuscript does a poor job of discussing previous studies.' (Reviewer 3)
We have taken this criticism seriously and applied several changes to the manuscript. To intensify the focus, the paper now concentrates clearly on the main question regard-

ing the variability of lower stratospheric age spectra on seasonal to inter-annual time scales (as suggested by Reviewer 1), and on the additional aspect regarding the effects of residual circulation and mixing on the spectrum from a global perspective. We removed 3 figures (Figs. 5, 10, 15 in the submitted version), two subsections of the discussion (Sects. 6.2, 6.3 in the submitted version), and the discussion of ENSO-related variability in Sect. 5 (suggested by Reviewer 1). The parts from the old Sect. 6.2, concerning variability of the fraction of young air, are now included in Sect. 3, which presents the results regarding seasonality. The revised discussion therefore clearly focusses on the seasonal and inter-annual variability in the age spectrum and how this variability generates the multiple peaks in the spectrum. Regarding inter-annual variability, we only discuss the QBO, as this is the dominant mode of inter-annual variability in the tropical lower stratosphere and its effects on the age spectrum have not been studied in detail, so far. The discussion of QBO-related effects in Sect. 5 is enhanced (as suggested by Reviewer 2) and an additional figure (new Fig. 12) is used, presenting easterly and westerly QBO composites, to further illustrate these effects (as suggested by Reviewer 3). A by-product of these changes is that the revised manuscript now is substantially shorter (as requested by Reviewer 2).

We changed large parts of the text and included 'strategically placed clarifying phrases', as required by Reviewer 1, in order to enhance clarity of the discussion. A new figure (Fig. 2) is introduced, showing the dispersal of the winter/summer tracer pulses, and is referred to in Sect. 2 and in the discussion to illustrate the transport processes discussed. In particular, we thank Reviewer 3 to point out certain passages in the text where the submitted manuscript was not precise about the existing literature. It was definitely not our intention to mix up results of this paper with what is already known. Hence, we carefully revised the manuscript in order to cite existing literature correctly and to clearly state what is known already and what is not.

General comment:

*This paper is very well written, enlightening, and concerns an important and timely aspect of atmospheric transport. My primary criticism is that the discussion is too terse in places. I suggest the addition of strategically placed clarifying phrases will help the reader understand the logic of the arguments. In the specific comments, I have noted a few examples where I had a particularly difficult time following the logic. In addition, the discussion of the relationship between ENSO and inter-annual variations of the age spectrum (Figure 10 and the last paragraph of Sec. 5) is not convincing. It should either be improved with some quantitative measures of the information content of that relationship or removed.*

Thank you for your encouraging comments. As mentioned in our 'General comment' above, we carefully revised the paper in order to enhance the clarity of the discussion ('strategically placed phrases', e.g. in introduction/paragraphs 4-5, discussion Sect. 6, text referring to new Fig. 2 in Sect. 2/6). To better focus the paper, the discussion of ENSO-related variability has been removed, as suggested.

Specific comments:

P. 2, line 17: *The discussion in this paragraph could greatly enhanced if you briefly explained (1) the motivation for representing transport as a diffusion process and that for representing the age spectrum by a Green's function, (2) the assumption implicit in those representations, and (3) why the assumptions needed for such a representation are not met.*

We rephrased the respective paragraph to be more precise about these points.

P. 2, line 34: *Changing 'using multiple pulses' to 'using multiple tracer pulses' will help clarify what you are doing.*

Done.

P. 6, line 16–19: *Perhaps I am confused here. Shouldn't the mean age calculated*

*from the age spectrum be identical to the value calculated from a perfectly linear clock tracer? Why is there a (small) discrepancy between the two calculations? Is it because, for ages approaching 10 years, the clock tracer does not capture the full range of source concentrations (i.e., the clock does not start early enough)?*

Thanks for pointing out this lack of clarity. Indeed, if the time span for convergence for both the clock-tracer and the age spectrum would be infinity both should be exactly equal. However, the calculations here effectively are of different lengths. We explain this point in detail in a new paragraph at the end of the appendix now (see also our reply to the 'Minor comment' of Reviewer 3).

P. 6, line 26–27: *It took me a while to decipher the meaning of the phrase 'coinciding with the youngest peak from spring to fall and the second youngest peak in winter'. Perhaps it would help to be more explicit, for example, 'the modal age as determined by the apex of the largest peak coincides with the youngest peak during spring, summer and fall and with the second peak during winter.'*

The respective sentence has been rephrased according to the Reviewer's suggestion.

P. 7, line 9–12: *The description of the propagation of the peak and the argument that the peaks are due to the fact that the most efficient transport from the boundary layer to the stratosphere is too terse. Perhaps another sentence or two will clarify the argument. For example, I can understand why efficient transport during winter leads to a peak in the summertime age spectrum at 6 months. However, I don't understand why the subsequent wintertime peaks are also linked to transport efficiency during winter. It seems to me that once the air is in the stratosphere (i.e., after 6 months) then an additional boundary layer to stratosphere transport boost the following winter is irrelevant; unless an important fraction of the air released during a specified winter remains in the troposphere until the following winter – when the troposphere-to-stratosphere 'transport window' re-opens.*

We admit that the wording was not clear. We changed the paragraph, and strongly

advise the reader of the discussion (Sect. 6), which has also been changed to enhance clarity (see our 'General comment' above).

P. 7, line 24–25: *Vertical velocity near the tropopause is also slower during summer than during winter (W. Randel and co-authors have written papers on this) and could be important.*

Indeed, the vertical velocity near the tropopause is slower during summer. However, particularly the reduced young air fractions in the summer hemisphere (Fig. 7 in the revised version) indicate that mixing plays an important role. Likewise, enhanced fractions of air older than 2 years directly above the tropopause (Fig. 7 c/d) can only be caused by in-mixing of old extratropical air. However, in order to better focus the manuscript we removed the respective figure (old Fig. 5) and discuss the related issues now with reference to Fig. 7, in a revised paragraph.

P. 10, line 8–20: *The connection between ENSO and inter-annual variations of the age spectra is, perhaps, believable, but is not convincingly demonstrated in Fig. 10. This could be due to the fact that, while equatorial convection patterns shift substantially as sea surface temperature patterns shift, variations of the average strength of tropical convection have relatively weak connections to ENSO. It seems the authors are trying to see make too much of patterns that appear in Fig. 10 – a problem that can arise when analysis rely too heavily on a qualitative comparison and the ability of the human eye to recognize patterns in a chaotic system (whether or not the patterns are meaningful). At the very least, Fig. 10 should be changed to make it easier to discern how well the age spectra are related to ENSO. It would be better to make the analysis more quantitative. For example, how much to the inter-annual variance of the age spectra is explained by a lag-relationship with ENSO variability? It might be best to simply remove Fig. 10 and the last paragraph of Sect. 5. Fig. 11: Did you create this plot for the global release experiment? (as opposed to 15S-15N) Does it look the same for both experiments? If so, or even if not, that is an important comparison to make.*

We admit that the presented relation between age spectrum and ENSO variability was too vague. We did some further analyses on that, like investigating the global release experiment which yields very similar results to the presented reference. However, in order to shorten and better focus the paper (as suggested by all Reviewers) we followed the advise of Reviewer 1 and removed the figure and the related text. The discussion of inter-annual variability now focusses on the QBO, as the main mode of inter-annual variability in the tropical lower stratosphere (see also our 'General comment').

P. 11, line 11–12: *Does the distribution in Fig. 11b mean that the particle pulses do not always have the same mass? That the rate of particle release varies? If so, how is this justified physically? Regardless, please explain this figure a bit more.*

The pulse release rate at the surface is not varying over the year. The figure shows the distribution of source times of the air sampled in the stratosphere. This distribution shows a strong peak at NH winter source times, although the release rate at the surface was the same as in summer. We changed the respective paragraph in the discussion to be clearer about this point in the revised version.

Sect. 6.1, last 3 paragraphs: *This discussion could use elaboration. Please be sure that, each time the effect of some phenomenon on peaks is mentioned, the explanation for how (or why) the effect is carried out is clear.*

We revised parts of the discussion to enhance clarity (see also our 'General comment'). Furthermore, reference to the new Fig. 2, showing the dispersal of winter/summer pulses explicitly, should help illustrating the processes described in the text.

P. 12, line 26–7: *Regarding 'This flushing . . . has implications for the chemical composition . . .'. Can you give an example?*

The flushing of the summertime lowermost stratosphere with young air is e.g. important for short-lived species and pollutants. However, in order to better focus and shorten the paper (suggested mainly by Reviewer 3) we removed this part of the

discussion (see also our 'General comment')..

Technical details:

P. 2 L. 16: *'Many studies of stratospheric age . . .'*

The wording has been changed.

P. 3 L. 2: *Change 'spectra from seasonal . . .' to 'specta on seasonal . . .'*

Done.

P. 8 L. 19: *'plays a dominant role' is an over-statement without further analysis. Change 'dominant' to 'important'. Better yet, use 'plays a more important role during this season than during NH winter'.*

Done.

P. 8 L. 21: *Change 'The isolation of tropical air through the subtropical transport barriers' to 'imposed by the subtropical transport barriers' (or some analogous change) to avoid the contradictory imagery invoked by the words 'isolation' and 'through'.*

The entire paragraph is changed in the revised version, and in particular the wording mentioned above.

P. 9 L. 3: *Change 'transit times above' to 'transit times longer than'.*

Done.

---

## Author Comment (AC2) · 31 May 2016

We thank all three reviewers for their careful considerations of the manuscript and their well thought-out comments. These certainly helped to significantly improve the paper. In the following, we address all comments and questions raised (Reviewer's comments in italics). Text changes in the manuscript are highlighted in color (except minor wording changes). The main concerns of the reviewers were:
(i) 'The manuscript is not very focused.' (Reviewer 3)
(ii) 'The paper is a bit long.' (Reviewer 2)
(iii) 'The discussion is too terse in places.' (Reviewer 1)
(iv) 'The manuscript does a poor job of discussing previous studies.' (Reviewer 3)
We have taken this criticism seriously and applied several changes to the manuscript. To intensify the focus, the paper now concentrates clearly on the main question regarding the variability of lower stratospheric age spectra on seasonal to inter-annual time scales (as suggested by Reviewer 1), and on the additional aspect regarding the effects of residual circulation and mixing on the spectrum from a global perspective. We removed 3 figures (Figs. 5, 10, 15 in the submitted version), two subsections of the discussion (Sects. 6.2, 6.3 in the submitted version), and the discussion of ENSO-related variability in Sect. 5 (suggested by Reviewer 1). The parts from the old Sect. 6.2, concerning variability of the fraction of young air, are now included in Sect. 3, which presents the results regarding seasonality. The revised discussion therefore clearly focusses on the seasonal and inter-annual variability in the age spectrum and how this variability generates the multiple peaks in the spectrum. Regarding inter-annual variability, we only discuss the QBO, as this is the dominant mode of inter-annual variability in the tropical lower stratosphere and its effects on the age spectrum have not been studied in detail, so far. The discussion of QBO-related effects in Sect. 5 is enhanced (as suggested by Reviewer 2) and an additional figure (new Fig. 12) is used, presenting easterly and westerly QBO composites, to further illustrate these effects (as suggested by Reviewer 3). A by-product of these changes is that the revised manuscript now is substantially shorter (as requested by Reviewer 2).

We changed large parts of the text and included 'strategically placed clarifying phrases', as required by Reviewer 1, in order to enhance clarity of the discussion. A new figure (Fig. 2) is introduced, showing the dispersal of the winter/summer tracer pulses, and is referred to in Sect. 2 and in the discussion to illustrate the transport processes discussed. In particular, we thank Reviewer 3 to point out certain passages in the text where the submitted manuscript was not precise about the existing literature. It was definitely not our intention to mix up results of this paper with what is already known. Hence, we carefully revised the manuscript in order to cite existing literature correctly and to clearly state what is known already and what is not.

General comment:

*This paper uses trajectories from the CLaMS transport model driven by ERA-Interim meteorology to analyze stratospheric age of air spectra. This work provides unique and revealing insights into the stratospheric transport on various time scales and latitude regions. The authors have recently done nice work on explaining various aspects of the stratospheric mean age of air and how the residual mean circulation and isentropic mixing contribute to the observed mean age distributions. But as they mention here, the age spectra provide another level of information, in particular how the transport variability imprints on the age spectra for years afterward. The analysis is excellent and I highly recommend publication with consideration of the minor comments below. The paper is a bit long, 16 figures is quite a lot but I don't have any specific suggestions on what could be left out. Perhaps with fewer figures a bit more time could be spent explaining some of the new and unique features of the remaining plots.*

Thank you for your encouraging comments. As explained in our 'General comment' above, we revised the manuscript in order to better focus and shorten it.

Minor comments:

P. 2, line 14: *awkward sentence, change to something like "comes with the benefit of allowing one to separate. . ."*

Done.

P. 2, line 35: *". . .and allows one to calculate the. . ."*

Done.

P. 2: *Just a comment that in my Ray et al. [2014] paper figure 4 shows age spectra from the TLP model with multiple peaks with clear seasonal and QBO influence. We didn't explain what caused the peaks beyond the known variability in the MERRA transport input to the model but thought you might want to include a mention of this paper here.*

We apologize for missing this reference here. A notice is included in the introduction of

the revised manuscript version.

P. 7, line 13: *". . .allows one to quantify air. . ."*

Done.

P. 8, line 13–15: *The younger tropical spectrum peak in DJF in Figure 6 is hard to see. I'll take your word for it but this figure doesn't seem to back up that statement unless I'm missing something.*

We admit that it is difficult to read this difference from the figure. We therefore explicitly state the numbers of the 20N/S average modal age in the text now (3.5 months during NH winter versus 4.1 months during summer).

P. 10, line 1–8: *Figure 9 is really nice and has so many features it takes some time to appreciate them all and what they mean. The propagation of the signals to older parts of the age spectra with time is reminiscent of the tropical tape recorder signal but has a different physical meaning. It's actually difficult to interpret physically what's going on with these signals because it's in a different phase space than we're used to thinking about. For instance, in Figure 9c-f there are anomalous peaks and troughs in the spectra that appear to propagate from the time when an event like a QBO easterly phase occurred at ages from 1-2 years for the following 4 years out to ages of 6 years. The actual air masses that were influenced by the QBO transport anomaly move around the stratosphere and many of them actually leave the stratosphere over the following 4 years and yet at this theta level and latitude range there is a signal that remains. It's as though the air influenced by the particular QBO event circulates around and enough of it comes back through this theta and latitude region to maintain an anomalous signal. That's actually remarkable! It might be worth spending a bit more time explaining the physical meaning of this plot and the features since I don't think it's obvious and I'm not even sure I'm getting the full picture.*

We significantly enhanced the discussion of the QBO-related inter-annual age spec-

[Figure]

trum variability in Sect. 5 (including the new Fig. 12 showing QBO composites), trying to thoroughly address all points raised by the Reviewer.

Figure 5: *in the caption the hemispheres are switched for (b) and (c).*

Yes - however, we removed this figure from the revised manuscript order to shorten the paper. The corresponding discussion now refers to Fig. 7.

Figure 6: *in the caption should be "show"*

Corrected.

Figure 7: *In the interest of shortening the paper could be one to consider removing.*

We kept this figure, but removed other figures as explained in our 'General comments'.

Figure 9: *Really interesting as mentioned above and a lot going on here. My suggestion to be able to see the features more clearly on b,d and f primarily is to separate out the delta age, mode and RCTT that are on the right axis into their own plots. It's hard to see their oscillation around the zero line as it is and it obscures somewhat the propagation of the pdf anomalies.*

We admit that the figure includes a lot of information. However, we think it is good to have the delta age, mode and RCTT and age spectra together in one figure to be able to relate them easily, without the need to compare different figures. As the modal age anomalies for the extratropics (Fig. 13 d/f) are characterized by particularly large scatter and contain no substantial additional information we removed these from the respective panels, following the Reviewer's advice. Furthermore, we enlarged Fig. 13 a bit to enhance readibility.

Figure 14: *Is each of the 3 lines of each color an individual month within DJF?*

Each line is the average over DJF, with different age spectra of the same color representing different latitudes within the respective latitude band (corresponding to the respective color). We state this explicitly in the caption now.

---

## Author Comment (AC3) · 31 May 2016

F. Ploeger and T. Birner

f.ploeger@fz-juelich.de

We thank all three reviewers for their careful considerations of the manuscript and their well thought-out comments. These certainly helped to significantly improve the paper. In the following, we address all comments and questions raised (Reviewer's comments in italics). Text changes in the manuscript are highlighted in color (except minor wording changes). The main concerns of the reviewers were:
(i) 'The manuscript is not very focused.' (Reviewer 3)
(ii) 'The paper is a bit long.' (Reviewer 2)
(iii) 'The discussion is too terse in places.' (Reviewer 1)
(iv) 'The manuscript does a poor job of discussing previous studies.' (Reviewer 3)
We have taken this criticism seriously and applied several changes to the manuscript. To intensify the focus, the paper now concentrates clearly on the main question regard-

ing the variability of lower stratospheric age spectra on seasonal to inter-annual time scales (as suggested by Reviewer 1), and on the additional aspect regarding the effects of residual circulation and mixing on the spectrum from a global perspective. We removed 3 figures (Figs. 5, 10, 15 in the submitted version), two subsections of the discussion (Sects. 6.2, 6.3 in the submitted version), and the discussion of ENSO-related variability in Sect. 5 (suggested by Reviewer 1). The parts from the old Sect. 6.2, concerning variability of the fraction of young air, are now included in Sect. 3, which presents the results regarding seasonality. The revised discussion therefore clearly focusses on the seasonal and inter-annual variability in the age spectrum and how this variability generates the multiple peaks in the spectrum. Regarding inter-annual variability, we only discuss the QBO, as this is the dominant mode of inter-annual variability in the tropical lower stratosphere and its effects on the age spectrum have not been studied in detail, so far. The discussion of QBO-related effects in Sect. 5 is enhanced (as suggested by Reviewer 2) and an additional figure (new Fig. 12) is used, presenting easterly and westerly QBO composites, to further illustrate these effects (as suggested by Reviewer 3). A by-product of these changes is that the revised manuscript now is substantially shorter (as requested by Reviewer 2).

We changed large parts of the text and included 'strategically placed clarifying phrases', as required by Reviewer 1, in order to enhance clarity of the discussion. A new figure (Fig. 2) is introduced, showing the dispersal of the winter/summer tracer pulses, and is referred to in Sect. 2 and in the discussion to illustrate the transport processes discussed. In particular, we thank Reviewer 3 to point out certain passages in the text where the submitted manuscript was not precise about the existing literature. It was definitely not our intention to mix up results of this paper with what is already known. Hence, we carefully revised the manuscript in order to cite existing literature correctly and to clearly state what is known already and what is not.

General comment:

*This manuscript presents calculations of the seasonal and inter annual variations in the stratospheric age spectrum obtained from the CLaMS model driven by ERA-Interim winds. The manuscript contains some interesting and new results that certainly warrant publication. However, the manuscript is not very well focused, does a poor job of discussing previous studies, and some of the major conclusions (as stated in abstract or conclusions) are not new results. I think the manuscript needs to be revised to be more focused on the new aspects of their calculations (seasonal and inter annual variability, including QBO) and to put these in context of previous studies.*

We appreciate the constructive criticism. As explained in our 'General comment' above, we significantly changed the paper (mainly introduction, Sects. 3, 5 and discussion) in order to better focus on seasonal and inter-annual variability, and in order to correctly cite the existing literature and to state what is known and what is new (see reply to the detailed concerns below).

Major comments:

1: *The manuscript is not very focused, and the new results are not clearly presented. This lack of focus can be seen from the title, which is very vague (and doesn't match much of the manuscript). In my opinion the relatively new and important aspect of the study is looking at seasonality and inter annual variability of the age spectrum, which has only been done in a few previous studies, and none of the previous studies have included the QBO (as the authors highlight in the Introduction). However there is actually relatively little discussion of these aspects, and there is just as much (or even more) discussion is on the age spectrum - mean age relationship, the approximation of G by inverse Gaussian, and comparisons of residual circulation with modal age. All of these latter issues could be examined using steady flows, and much of what is discussed is already known. My recommendation is to focus on the seasonal and inter annual variations (and QBO) and to minimize the discussion of the other issues.*

The manuscript is significantly changed to focus more clearly on seasonality and inter annual variability of the age spectrum, as suggested (see also our 'General comment'). We also changed the title to be more precise about that ('Seasonal and interannual variability of lower stratospheric age of air spectra'). The discussion of the approximation of G by an inverse Gaussian has been removed, and the main message condensed into the last sentence of the discussion. The discussion subsection on the age spectrum - mean age relationship has also been removed. The seasonality of young air fractions (which was previously included in this discussion part) is now presented in the seasonality section 3.

2: *While there are references to previous studies that examined similar aspects of the age spectrum in the Introduction there is very little discussion of these when interpreting the calculations presented here. Because of this it is not clear to me how much of the results are new and how much are just reproducing earlier results with Lagrangian model driven by reanalyses. The clearest examples of lack of discussion of previous studies are Sections 3 (seasonality) and 5 (inter annual variability) where there is not a single mention of the Li et al. 2012a,b studies which examined exactly these issues. How do the results presented compare with these previous studies? What is new in what is presented (other than a slightly different approach)?*

Thank you for the constructive criticism! We have tried to improve the discussion of previous literature throughout the text. Examples are:
(i) The introduction, where we now explain in detail that most existing age spectrum studies are based on the assumption of a stationary flow, and that only a few recent ones considered time-dependent age spectra (e.g., Reithmeier et al., 2007; Diallo et al., 2012; Li et al., 2012a/b). We frequently refer to these studies in the introduction, in Sect. 2 and in the discussion. Furthermore, in the revised version we also explain at the beginning of Sect. 3 that the occurrence of multiple peaks in the age spectrum has already been pointed out in recent publications, and we discuss probable causes in the discussion (Sect. 6) in relation to the literature (e.g., Reithmeier et al., 2007;

Diallo et al., 2012; Li et al., 2012a/b). In the discussion, we refer first (P13, L9ff) to the discussions of the peaks in these papers, explain that there is no common understanding, and finally discuss the CLaMS results in relation to them. The fact that in the reanalysis-driven CLaMS simulation multiple spectrum peaks emerge not only at high latitudes but almost globally is clearly different to previous studies, and thoroughly discussed now. As further discussed, we think that it is neither of the recently proposed mechanisms which causes these peaks, but a combination of them (see discussion).

In particular, our analysis is the first full transport model analysis on age spectra based on reanalysis winds including a QBO. Such inter-annual variability can be easily investigated using our modified BIER approach, as it is based on a transient multi-year simulation (Li et al., 2012a/b for instance considered only time-slice experiments). The clear effect of the QBO in modifying and even generating age spectrum peaks is clearly a new result.

Furthermore, in the introduction we now explain precisely (P3, L12ff) that in the tropics the modal age is known to be closely related to the residual circulation, but that no global investigations exist on that, to our knowledge. Hence, our global analysis on regions and seasons where modal age is controlled by the residual circulation (the tropics and the wintertime extratropical stratosphere above about 500 K) is, in our opinion, also a new result.

3: *Some of the conclusions (as stated in the abstract) are well known facts. This paper may be providing more support, but as written it appears these are new results. One example is the statement in the abstract that "Interpretation of the age spectrum in terms of transport contributions due to the residual circulation and mixing is generally not straightforward." This is well known and not sure this counts as a significant enough statement for an abstract. Another example are the statements towards the end of both the abstract and conclusions regarding benefits of age spectrum calculations and need for inclusion in model inter comparisons. Again not new, and in fact age spectrum calculations were actually included in model inter-comparisons in late 1990s (Hall et*

*al. 1999). Is this really the major take home message from this manuscript (ending abstract and conclusions with this gives this impression)?*

As stated above (see our 'General comments'), we tried to better focus the paper on seasonal and inter-annual age spectrum variations. We also removed two sentences from the abstract to be more focussed here. We kept the last part of the conclusions, although it is indeed not new that considering the age spectrum is beneficial compared to mean age (the full skewed spectrum always includes more information than just its first moment). We included citations of two existing examplary papers on that (Hall, 1999; Waugh and Hall, 2002). However, time-dependent age spectrum diagnostics have not been implemented in global models as standard transport diagnostics, hitherto. While advantages of considering the full age spectrum have certainly been stressed many times before, most studies in the last ~10 years focus still on mean age, which is a quantity very dangerous to interprete. We therefore feel that it can't hurt to (re-)stress the benefits of age spectra analyses, and to conclude the paper with a related (slightly modified) paragraph.

4: *There are multiple places where I think previous papers or the current understanding are misrepresented. Examples:*

P. 1, line 24: *As written it suggests that the Hall and Plumb 1994 and Waugh and Hall 2002 papers did not appreciate that there was a range of pathways and an age spectrum, whereas the opposite is true and Hall and Plumb focused on this fact.*

This misrepresentation was certainly not intended - we fully agree about the merits of these previous studies. We placed the respective citation to a different place not to cause confusion.

P. 2, line 6: *I don't think it is correct to indicate that the apparent disagreement between observed and models changes in transport is due to age spectrum versus mean age differences. This disagreement occurs if you compare mean age from observations with mean age from models, so not a residual circulation vrs mean age issue.*

Agreed, although the representation of mixing, and in particular corresponding long-term changes, is likely less robust across models. We have restructured this paragraph somewhat, which should emphasize the structural BDC changes more.

P. 2, line 28: *Is it a common view that modal age can be related to residual mean mass circulation? Maybe in the tropical lower stratosphere, but I don't it is common to think such a relationship extends beyond this region.*

We admit that we were imprecise with this statement. In the tropics this relation between the mode and residual circulation is known. However, globally no study on this relation exists to our knowledge. Therefore, our results confirm the common view in the tropics and further show that a similar relation holds in the wintertime extratropical stratosphere above about 500 K, whereas particularly in the summertime lower stratosphere mixing effects are important. We changed the text at several places in order to be precise (e.g., P3, L12ff).

P. 13, line 19: *It is not correct to say that "Hall and Plumb (1994) argued that the stratospheric age spectrum may be well approximated by the Green's function for a one-dimensional diffusion process". They used the 1 diffusion model to illustrate aspects of the age spectrum not to model the actual stratospheric age spectrum (they are explicit about this: "our goal at this point is not to quantify stratospheric transport, but rather to illustrate the points of the previous discussion").*

Thanks for pointing the confusing statement out! We entirely agree with the Reviewer's view and changed the text accordingly (corresponding text changes at two places in the revised manuscript: introduction/L20ff, and first paragraph of Sect. 3).

P. 14, line 7: *"This process can evidently not be described by simple one-dimensional diffusion." I think this well known, not only from looking at G(t) from other threedimensional models, but also from the tropical leaky pipe model where expressions for the age spectrum have been derived (Hall 2000) (see also Waugh and Hall (2002) review).*

We agree with the Reviewer. However, the entire discussion part about age spectrum fits has been removed from the revised manuscript, in order to better focus the paper (as suggested by the Reviewer, see also out 'General comments').

Minor comments:

P. 6, line 15–19: *How long was the clock tracer run for; is it in (quasi-)steady state? It is stated that the agree is good but in polar regions the clock tracer is younger by over 0.5 yrs. Note, the paper of Hall and Haine (2002) might be relevant here. They derive the relationship between "ideal age" (which is an alternate clock tracer) and "mean of age spectrum". In their example the clock tracer converges quicker than the mean of the age spectrum, which appears opposite to your result. However, this may because you have run calculations for different length.*

Thanks for pointing out this lack of clarity. Indeed, the calculations for the two quantities effectively have a different length. The clock-tracer was subject to a 10 year spin-up (repeating 1979 conditions) and another 10 years of transient simulation (1979-1988) before taking into consideration, whereas the age spectrum has explicitly been calculated over 10 years of transit time (1979-1988 simulation) but with the tail fitted to infinite transit times. Hence, the effective calculation length is longer for the corrected age spectrum than for the 'clock-tracer', and there is no contradiction to the results of (Hall and Holzer, 2002). This is explained in detail now in a new paragraph at the end of the appendix (see also our reply to Reviewer 1).

Eq. 7: *Why reference Bonisch et al 2009? As you have just mentioned the G(t) for 1D advection diffusion was used in Hall and Plumb, and this same expression was given in the Waugh and Hall (2002) review paper. Furthermore, this is not a parameterization, this is the exact G(t) just expressed in terms of the mean and width (rather than flow velocity and diffusion).*

We changed the text according to the Reviewer's suggestion, and refer to Waugh and

Hall (2002) in the revised paper.

Section 5: *Is it possible to make composites on east and west phase of the QBO?*

Thanks for this suggestion. We present composites for easterly and westerly QBO phases now in the new Fig. 12 at the beginning of Sect. 5 (inter-annual variability). These composites clearly illustrate how the fraction of young air is increased during QBO easterly phase, due to anomalously strong and fast tropical upwelling, and how the spectrum tail becomes more pronounced during QBO westerly phase. Remakably, secondary peaks develop in the tropical age spectrum at older transit times during QBO westerly phase, indicating an increased impact of in-mixing of old extratropical air on tropical composition. We discuss these findings in Sect. 5, and enhanced the discussion about the QBO-related age spectrum variability.

P. 11, line 16: *Are the results in Orbe et al. (2014) relevant for this discussion?*

Indeed, the fact that pulses released during summer are more efficiently dispersed meridionally before they reach the tropical pipe and therefore are less likely to undergo recirculation in the stratosphere would be consistent with a stronger return flux into the troposphere for air entering the stratosphere in July compared to January, as found by Orbe et al. (2014). We mention this in the revised discussion version now. However, it is not clear how much this comparison really tells because of the different pulse tracer settings (this paper: released at the surface every other month; Orbe et al.: released at the thermal tropopause in January and July)